# OxonFair: A Flexible Toolkit for Algorithmic Fairness

**Eoin Delaney**
University of Oxford

**Zihao Fu**
University of Oxford

**Sandra Wachter**
University of Oxford

**Brent Mittelstadt**
University of Oxford
`firstname.lastname@oii.ox.ac.uk`

**Chris Russell**
University of Oxford

## Abstract

We present OxonFair, a new open source toolkit for enforcing fairness in binary classification. Compared to existing toolkits: (i) We support NLP and Computer Vision classification as well as standard tabular problems. (ii) We support enforcing fairness on validation data, making us robust to a wide range of overfitting challenges. (iii) Our approach can optimize any measure based on True Positives, False Positive, False Negatives, and True Negatives. This makes it easily extensible and much more expressive than existing toolkits. It supports all 9 and all 10 of the decision-based group metrics of two popular review articles. (iv) We jointly optimize a performance objective alongside fairness constraints. This minimizes degradation while enforcing fairness, and even improves the performance of inadequately tuned unfair baselines. OxonFair is compatible with standard ML toolkits, including sklearn, Autogluon, and PyTorch and is available at `https://github.com/oxfordinternetinstitute/oxonfair`.

## 1 Introduction

The deployment of machine learning systems that make decisions about people offers an opportunity to create systems that work for everyone. However, such systems can lock in existing prejudices. Limited data for underrepresented groups can result in ML systems that do not work for them, while the use of training labels based on historical data can result in ML systems copying previous biases. As such, it is unsurprising that AI systems have repeatedly exhibited unwanted biases towards certain demographic groups in a wide range of domains including medicine [1, 2], finance [3, 4], and policing [5]. Such groups are typically identified with respect to legally protected attributes, such as ethnicity or gender [6, 7, 3]. The field of algorithmic fairness has sprung up in response to these biases.

Contributions to algorithmic fairness can broadly be split into methodological and policy-based approaches. While much methodological work focuses on measuring and enforcing (un)fairness, a common criticism from the policy side is that this work can occur *'in isolation from policy and civil societal contexts and lacks serious engagement with philosophical, political, legal and economic theories of equality and distributive justice'* [8].

In response to these criticisms, we have developed OxonFair, a more expressive toolkit for algorithmic fairness. We acknowledge that people designing algorithms are not always the right people to decide on policy, and as such we have chosen to create as flexible a toolkit as possible to allow policymakers and data scientists with domain knowledge to identify relevant harms and directly alter the system behaviour to address them. Unlike existing Fairness toolkits such as AIF360 [9], which take a method-driven approach, and provide access to a wide range of methods but with limited control over their behaviour, we take a measure-based approach and provide one fairness method that is extremely customizable, and can optimize user-provided objectives and group fairness constraints.

38th Conference on Neural Information Processing Systems (NeurIPS 2024).

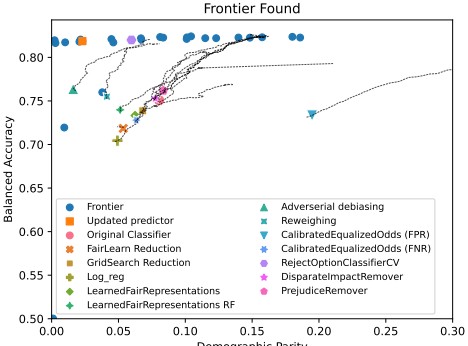

| Classifier (dataset) | Partition | Fairlearn | | OxonFair | |
|---|---|---|---|---|---|
| | | Acc (↑) | DEO (↓) | Acc (↑) | DEO (↓) |
| Decision Tree | Train/Val | 100% | 0% | 82% | 2.0% |
| (adult) | Test | 81% | 8.8% | 81% | 1.1% |
| Random Forest | Train/Val | 100% | 0% | 86% | 1.6% |
| (adult) | Test | 86% | 7.5% | 86% | 3.3% |
| XGBoost | Train/Val | 100% | 0% | 90% | 0.6% |
| (myocardial infarction) | Test | 89% | 11.8% | 87% | 2.9% |

| Criterion | AIF360 | Fairlearn | OxonFair |
|---|---|---|---|
| Number of methods | 10+ | 5 | 1 |
| Adjustible Fairness Criteria | ✗ | ✓ | ✓ |
| Supports 3+ Groups | ✗ | ✓ | ✓ |
| Fairness definitions enforced per method | <4 | 5 | 14+ |
| Methods needing groups at eval | Some | 1 | No |
| Supports Utility Functions | ✗ | ✗ | ✓ |
| Supports Tabular Data | ✓ | ✓ | ✓ |
| Supports Computer Vision | ✗ | ✗ | ✓ |
| Supports NLP | ✗ | ✗ | ✓ |

Figure 1: **Left:** The need for an objective when enforcing fairness. We evaluate a range of methods with respect to balanced accuracy and demographic parity (OxonFair generates a frontier of solutions). Only OxonFair and RejectOptimization optimize balanced accuracy. As we improve the balanced accuracy of fair methods by adjusting classification thresholds (gray lines) fairness deteriorates. To avoid this, we jointly optimize a fairness measure and an objective. For more examples, see Figure 6. **Right Top:** Using validation data in fairness. We compare against Fairlearn using standard algorithms with default parameters. These methods perfectly overfit and show no unfairness with respect to equal opportunity on the trainset, but substantial unfairness on test. OxonFair enforces fairness on held-out validation data and is less prone to overfitting. **Right Bottom:** A comparison of toolkits. AIF360 offers a large range of tabular methods, most of which do not allow fairness metric selection, Fairlearn offers fewer but more customizable tabular methods. OxonFair offers one method that can be applied to text, image, and tabular data, while supporting more notions of fairness and objectives.

To do this, we focus on one of the oldest and simplest approaches to group fairness: per-group thresholding [10, 11, 3], which is known to be optimal for certain metrics under a range of assumptions [3, 12, 13]. Our contribution is to make this as expressive as possible while retaining speed, for the relatively low number of groups common in algorithmic fairness. Inherently, any approach that allows a sufficiently wide set of objectives, and sets per-group thresholds will be exponential with respect to the number of groups, but we use a standard trick, widely used in the computation of measures such as mean absolute precision (mAP) to make this search as efficient as possible. Accepting the exponential complexity allows us to solve a much wider range of objectives than other toolkits, including maximizing F1 or balanced accuracy (see Figure 1 left), minimizing difference in precision [14], and guaranteeing that the recall is above k% for every group [8]. Where groups are unavailable at test time, we simply use a secondary classifier to estimate group memberships [15, 16] and set different thresholds per inferred group to enforce fairness with respect to the true groups.

Thresholding can be applied to most pretrained ML algorithms, and optimal thresholds can be selected using held-out validation data unused in training. This is vital for tasks involving deep networks such as NLP and computer vision, where the training error often goes to zero, and fairness methods that balance error rates between groups cannot generalize from constraints enforced on overfitting training data to previously unseen test data [17]. While overfitting is unavoidable in vision and NLP tasks, it is still a concern on tabular data. Figure 1 Top-Right, shows examples of decision trees, random forests [18] and XGBoost [19] trained with default parameters and obtaining 0 training error on standard datasets. This causes the Fairlearn reductions method [20] to fail to enforce fairness.

NLP and vision are so challenging that two popular toolkits Fairlearn and AIF360 do not attempt to work in these domains. In contrast, we target them, making use of a recent work [21] that showed how fair classifiers based on inferred group thresholds can be compressed into a single network.

## 2 Related Work

Bias mitigation strategies for classification have been broadly categorized into three categories [6, 22–24]; pre-processing, in-processing and post-processing.

**Pre-processing** algorithms improve fairness by altering the dataset in an attempt to remove biases such as disparate impact [11] before training a model. Popular preprocessing approaches include simply reweighting samples in the training data to enhance fairness [25], optimizing this process by learning probabilistic transformations [26], or by generating synthetic data [27–29].

**In-processing / In-training** methods mitigate bias by adjusting the training procedure. Augmenting the loss with fair regularizers [23, 30] is common for logistic regression and neural networks. Agarwal et al. [31] iteratively alter the cost for different datapoints to enforce fairness on the train set. Approaches based on adversarial training typically learn an embedding that reduces an adversary's ability to recover protected groups whilst maximizing predictive performance [32–35]. Other popular approaches include Disentanglement [36, 37], Domain Generalization [38–40], Domain-Independence [41] and simple approaches such as up-sampling or reweighing minority groups during training. Notably, in the case of high-capacity models in medical computer-vision tasks, a recent benchmark paper by Zong et al. [42] showed state-of-the-art in-processing methods do not significantly improve outcomes over training without consideration of fairness. A comprehensive benchmark study of in-processing methods in other domains is provided by Han et al. [43].

**Post-processing** methods enforce fairness by using thresholds and randomization to adjust the predictions of a trained model based on the protected attributes [3, 44]. Post-processing methods are typically *model-agnostic* and can be applied to any model that returns confidence scores.

**Enforcing Fairness on Validation Data** avoids the misestimation of error rates due to overfitting. It has shown particular promise in computer vision through Neural Architecture Search [45], adjusting decision boundaries [30], reweighting [46] and data augmentation [17].

## 2.1   Fairness Toolkits

Most toolkits such as Fairness Measures [47], TensorFlow Fairness Indicators [48], and FAT Forensics [49] focus on measuring bias and do not support enforcing fairness through bias mitigation. FairML [50] audits fairness by quantifying the importance of different attributes in prediction. This is best suited for tabular data where features are well-defined. FairTest [51] investigates the associations between application outcomes (e.g., insurance premiums) and sensitive attributes such as age to highlight and debug bias in deployed systems. Aequitas [52] provides examples of when different measures are (in)appropriate with support for some bias mitigation methods in binary classification. Themis-ML [53] supports the deployment of several simple bias mitigation methods such as relabelling [25], but focuses on linear models. Friedler et al. [22] introduce the more complete Fairness Comparison toolkit where four bias mitigation strategies are compared across five tabular datasets and multiple models (Decision trees, Gaussian Naïve Bayes, SVM, and Logistic Regression).

There are two fairness toolkits that support sklearn [18] like OxonFair. These are the two most popular toolkits: Microsoft Fairlearn [20] (1.9k GitHub Stargazers as of June 2024) and IBM AIF360 [9] (2.4k Stargazers). AIF360 offers a diverse selection of bias measures and pre-processing, in-processing and post-processing bias mitigation strategies on binary classification tabular datasets. For mitigation, Fairlearn primarily offers implementations of [31, 3] avoiding the use of the term *bias*, instead considering fairness through the lens of fairness-related harms [54] where the goal is to *"help practitioners assess fairness-related harms, review the impacts of different mitigation strategies and make trade-offs appropriate to their scenario"*. Lee & Singh [55] recognized Fairlearn as one of the most user-friendly fairness toolkits, and critiqued AIF360 as being the least user-friendly toolkit.

Both AIF360 and Fairlearn contain post-processing methods that select per-group thresholds. Unlike OxonFair, neither method uses the fast optimization we propose; both methods require group information at test time; AIF360 only supports two groups, but does use cross-validation to avoid overfitting; Fairlearn does not support the use of validation data, but does support more than two groups. According to their documentation, neither toolkit can be applied to NLP or computer vision.

**Specialist solvers**   Fairret [56] is a recent PyTorch library shown to enforce fairness on tabular data. As PyTorch is a focus of OxonFair (see Section 4.2), we compare with Fairret in Appendix D.1. Cruz and Hardt [57] proposed an efficient LP-based formulation for post-processing Equalized Odds. It supports randomized thresholds, and dominates fairness methods such as OxonFair that use only one threshold per group. In Appendix D.2 we show how OxonFair can be extended to support randomized

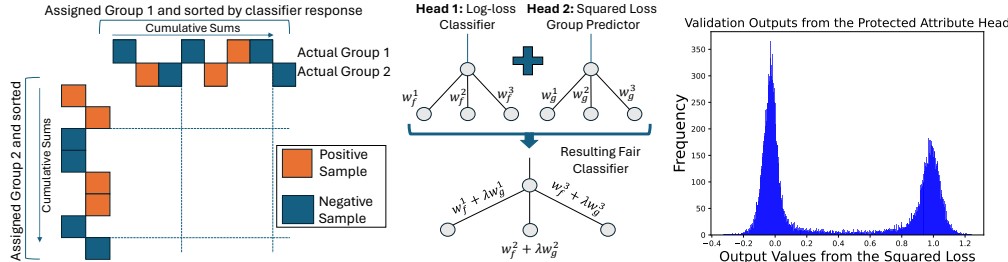

Figure 2: **Left:** Summary of the fast path algorithm for inferred attributes (Section 4.1). Groups are noisily estimated using a classifier. Within each estimated group, we cumulatively sum positive and negative samples that truly belong to each group. For each pair of thresholds, we select relevant sums from the inferred group and combine them. See Appendix A.1. **Center:** Combining two heads (original classifier and group predictor) to create a fair classifier. See Section 4.2. **Right:** The output of a second head predicting the protected attribute in CelebA. The pronounced bimodal distribution makes the weighted sum of the two heads a close replacement for per-group thresholds.

thresholds, alongside a determenistic variant and another using inferred group membership that are not supported by [57].

## 3 Toolkit interface

The interface of OxonFair decomposes into three parts: (i) evaluation of fairness and performance for generic classifier outputs. (ii) evaluating and enforcing fairness for particular classifiers. (iii) specialist code for evaluating and enforcing fairness for deep networks.

Code for the evaluation of classifier outputs takes target labels, classifier outputs, groups, and an optional conditioning factor as input; while code for the evaluation and enforcement of fairness of a particular classifier, are initialized using the classifier, and from then on take datasets (in the form of a pandas dataframe [58], or a dictionary) as input, and automatically extracts these factors from them.

The evaluation code provide three functions: `evaluate` which reports overall performance of the classifier; `evaluate_per_group` which reports performance per group of the classifier; and `evaluate_fairness` which reports standard fairness metrics. All methods allow the user to specify which metrics should be reported. We recommend data scientists focus on `evaluate_per_group` which shows direct harms such as poor accuracy, precision, or low selection rate for particular groups.

OxonFair provides an interface `FairPredictor(classifier, validation_data, groups)` that takes an existing classifier as an input, a validation dataset, and specification of the groups as an input and returns an object which we then enforce fairness on by calling `.fit(objective, constraint, value)`. Internally, the method explores a wide range of possible thresholds for each group, membership of which is assumed to be either known or inferred by an auxiliary classifier.

The resulting `FairPredictor` has evaluation methods as described above. When called without arguments, they report both the performance of the original and the updated fair classifier on the validation data. In addition, `FairPredictor` provides methods `predict` and `predict_proba` which make fair predictions and return scores corresponding to the left-hand side of Equation (1).

Calling `fit` optimizes the objective – typically a relevant performance criteria such as accuracy, subject to the requirement that the constraint is either greater or less than the value. If the objective should be minimized or maximized is inferred automatically, as is the requirement that the constraint is less than or greater than the value, but this default behavior can be user overridden.

This is a relatively minimal interface, but one that is surprisingly expressive. By explicitly optimizing an objective, we can not just minimize the degradation of the metric as we enforce fairness, but sometimes also improve performance over the unfair baseline that is not fully optimized with respect to this metric. Even when optimizing for accuracy, this can create situations where it looks like some improvements in fairness can be had for free, although generally this is an artifact of the gap between optimizing log-loss and true accuracy in training. By formulating the problem as a generic constrained optimization, and not requiring the constraint to be a typical fairness constraint, we leave it open

for enforcing a much broader space of possible objectives. This can be seen in Appendix C, where we show how to enforce minimax fairness [59], maximize utility [60] combined with global recall constraints, and demonstrate levelling-up [8] by specifying minimum acceptable harm thresholds.

Under the hood, a call to `fit` generates a Pareto frontier[1] and selects the solution that best optimizes the objective while satisfying the constraint. The frontier can be visualized with `plot_frontier`.

## 4 Inference

To make decisions, we assign thresholds to groups. We write $f(x)$ for the response of a classifier $f$, on datapoint $x$, $t$ for the vector corresponding to the ordered set of thresholds, and $G(x)$ for the one-hot encoding of group membership. We make a positive decision if

$$f(x) - t \cdot G(x) \geq 0 \tag{1}$$

To optimize arbitrary measures we perform a grid search over the choices of threshold, $t$.

**Efficient grid sampling** We make use of a common trick for efficiently computing measures such as precision and recall for a range of thresholds. This trick is widely used without discussion for efficient computation of the area under ROC curves, and we have had trouble tracking down an original reference for it. As one example, it is used by scikit-learn [18].

The trick is as follows: sort the datapoints by classifier response, then generate a cumulative sum of the number of positive datapoints and the number of negatives, going from greatest response to least. When picking a threshold between points $i$ and $i + 1$, TP is given by the cumulative sum of positives in the decreasing direction up to and including $i$; FN is the sum of negatives in the same direction; FP is the total sum of positives minus TP, and TN is the total sum of negatives minus TN.

We perform this trick per group, and efficiently extract the TP, FN, FP and TN for different thresholds. These are combinatorially combined across the groups and the measures computed. This two stage decoupling offers a substantial speed-up. If we write $T$ for the number of thresholds, $k$ for the number of groups, and $n$ for the total number of datapoints, our procedure is upper-bounded by $O(T^k + n \log n)$, while the naïve approach is $O(nT^k)$. No other fairness method makes use of this, and in particular, all the threshold-based methods offered by AIF360 make use of a naïve grid search.

From the grid sampling, we extract a Pareto frontier with respect to the two measures[2]. The thresholds that best optimize the objective while satisfying the constraint are returned as a solution. If no such threshold exists, we return the thresholds closest to satisfying the constraint.

### 4.1 Inferred characteristics

When using inferred characteristics, we offer two pathways for handling estimated group membership. The first pathway we consider makes a hard assignment of individuals to groups, based on a classifier response. The second pathway explicitly uses the classifier confidence as part of a per-datapoint threshold. In practice, we find little difference between the two approaches, but the hard assignment to groups is substantially more efficient and therefore allows for a finer grid search and generally better performance. However, the soft assignment remains useful for the integration of our method with neural networks, where we explicitly merge two heads (a classifier and a group predictor) of a neural network to arrive at a single fair model. For details of the two pathways see Appendix A.

### 4.2 Fairness for Deep Networks

We use the method proposed in [21] (N.B., they used it only for demographic parity). Consider a network with two heads $f$, and $g$, comprised of single linear layers, and trained to optimize two tasks on a common backbone $B$. Let $f$ be a standard classifier trained to maximize some notion of performance such as log-loss and $g$ is a classifier trained to minimize the squared loss[3] with respect to

---

[1]A maximal set of solutions such that for every element in the set, any solution with a better score with respect to the objective would have a worse score with respect to the constraint, and vice versa.

[2]In practice this is done twice, once in a coarse grid search to determine a good range, and then in a second finer search within the minimum and maximum range found by the first search

[3]The squared loss is used rather than log-loss so that the output of $g(x)$ remains close to 0 and 1. With log-loss, the output pre-sigmoid is more likely to overwhelm confident decisions made by the original classifier.

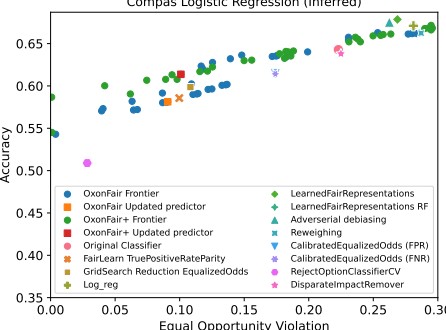

| | #Groups | Accuracy (↑) | Dem. Par. (↓) | Time (↓) |
|---|---|---|---|---|
| FairLearn 5 | | 86.5% | 3.8% | 47.0s |
| OxonFair 5 | | 86.9% | 1.9% | 0.6+ 42.9s |
| OxonFair(S) 5 | | - | - | - |
| FairLearn 4 | | 86.7% | 1.6% | 28.4s |
| OxonFair 4 | | 86.8% | 1.2% | 0.6+0.79s |
| OxonFair(S) 4 | | - | - | 0.6+411s |
| FairLearn 3 | | 86.5% | 0.7% | 25.0s |
| OxonFair 3 | | 86.8% | 2.1% | 0.6+ 0.07s |
| OxonFair(S) 3 | | - | - | 0.6+ 22.4s |
| FairLearn 2 | | 86.9% | 0.2% | 20.0s |
| OxonFair 2 | | 86.9% | 0.3% | 0.6+0.04s |
| OxonFair(S) 2 | | - | - | 0.6+1.2s |

Figure 3: **Left:** Results on Compas without using group annotations at test time. **Right:** Runtime Comparison for Fairlearn Reductions and OxonFair on Adult using a Macbook M2. To alter the groups, we iteratively merge the smallest racial group with 'Other', reducing the search space. For both methods, we enforced demographic parity over a train set consisting of 70% of the data. Despite the exponential complexity of our approach, we remain significantly faster until we reach 5 groups. The 0.6+ indicates the seconds to train XGBoost. OxonFair(S) indicates the runtime of the naive slow pathway described in Appendix A.2 rather than our accelerated approach.

a vector that is a one hot-encoding of group membership. Any decision $f(x) - t \cdot g(x) \geq 0$ can now be optimized for given criteria by tuning weights $w$ using the process outlined in the slow pathway.

As both $f$ and $g$ are linear layers on top of a common nonlinear backbone $B$, we can write them as:

$$f(x) = w_f \cdot B(x) + b_f, \quad g(x) = w_g \cdot B(x) + b_g \tag{2}$$

note that as $f(x)$ is a real number, and $g(x)$ is a vector, $w_f$ is a vector and $b_f$ a real number, while $w_g$ is a matrix and $b_g$ a vector.

This means that the decision function $f(x) - t \cdot g(x) \geq 0$ can be rewritten using the identity:

$$f(x) - t \cdot g(x) = w_f \cdot B(x) + b_f - t \cdot w_g \cdot B(x) - t \cdot b_g \tag{3}$$
$$= (w_f - t \cdot w_g) \cdot B(x) + (b_f - t \cdot b_g) \tag{4}$$

This gives a 3 stage process for enforcing any of these decision/fairness criteria for deep networks.

1. Train a multitask neural network as described above.
2. Compute the optimal thresholds $t$ on held-out validation data as described in Appendix A.
3. Replace the multitask head with a neuron with weights $(w_f - t \cdot w_g)$ and bias $(b_f - t \cdot b_g)$.

To maximize performance, the training set should be augmented following best practices, while, to ensure fairness, the validation set should not[4]. The resulting network $f^*$ will have the same architecture as the original non-multitask network, while satisfying chosen criteria.

OxonFair has a distinct interface for deep learning[5]. Training and evaluating NLP and vision frequently involves complex pipelines. To maximize applicability, we assume that the user has trained a two-headed network as described above, and evaluated on a validation set. Our constructor `DeepFairPredictor` takes as an input: the output of the two-headed network over the validation set; the ground-truth labels; and the groups. `fit` and the evaluation functionality can then be called in the same way. Once a solution is selected, the method `merge_heads_pytorch` generates the merged head, while `extract_coefficients` can be called to extract the thresholds $t$ from 4, when working with a different framework.

### 4.3   Toolkit expressiveness

Out of the box, OxonFair supports all 9 of the decision-based group fairness measures defined by Verma and Rubin [61] and all 10 of the fairness measures from Sagemaker Clarify [62]. OxonFair

---

[4]In some situations, a credible case can be made for including the mildest forms of augmentation, such as left-right flipping in the validation set, to maximize the validation set size.

[5]We provide example notebooks for practitioners to get started with `DeepFairPredictor` in our toolkit.

Table 1: We report mean scores over the 14 gender independent CelebA labels [28]. Single task methods and FairMixup scores in the second and third blocks are from Zietlow et al. [17]. ERM is the baseline architecture run without fairness. OxonFair (optimizing for accuracy and difference in equal opportunity (DEO)), has better accuracy (↑) and DEO (↓) scores than any other fair method.

| | ERM multitask | Uniconf. Adv.[74] | Domain Disc. [75, 41] | Domain Ind. [41] | OxonFair DEO | ERM single task | Debiasing GAN [28] | Regularized [76, 77] | g-SMOTE Adaptive [17] | g-SMOTE [17] | ERM [78] | FairMixup [78] |
|---|---|---|---|---|---|---|---|---|---|---|---|---|
| Acc. | **93.07** | 92.71 | 92.96 | 92.63 | 92.75 | 92.47 | 92.12 | 91.05 | 92.56 | **92.64** | **92.74** | 88.46 |
| DEO | 16.47 | 19.63 | 14.61 | 7.78 | **3.21** | 12.54 | 9.11 | **3.77** | 14.28 | 15.11 | 7.97 | **3.58** |

supports any fairness measure (including conditional fairness measures) that can be expressed per group as a weighted sum of True Positives, False Positives, True Negatives and False Negatives. OxonFair does not support notions of individual fairness such as fairness through awareness [63] or counterfactual fairness [64, 65].

See Appendix B for a discussion of how metrics are implemented and comparison with two review papers. Appendix C contains details of non-standard fairness metrics, including utility optimization [60]; minimax fairness [59, 66, 67]; minimum rate constraints [8], and Conditional Demographic Parity [68]. This also includes a variant of Bias Amplification [69, 70].

## 5   Experimental Analysis

For tabular data, we compare with all group fairness methods offered by AIF360, and the reductions approach of Fairlearn. OxonFair is compatible with any learner with an implementation of the method `predict_proba` consistent with scikit-learn [18] including AutoGluon [71] and XGBoost [19]. A comparison with Fairlearn and the group methods from AIF360 on the adult dataset can be seen in Figures 1 and 6 using random forests. This follows the setup of [9]: we enforce fairness with respect to race and binarize the attribute to white vs everyone else (this is required to compare with AIF360), 50% train data, 20% validation, and 30% test, and a minimum leaf size of 20. With this large leaf size, all errors on train, validation, and test are broadly comparable, but our approach of directly optimizing an objective and a fairness measure leads us to outperform others.

Figure 1 top right shows the importance of being able to use a validation set to balance errors. Using sklearn's default parameters we overfit to adult, and as the classifier is perfect on the training set, all fairness metrics that match error rates are trivially satisfied [72, 17]. The same behavior can be observed using XGBoost on the medical dataset [73] when enforcing equal opportunity with respect to sex[6]. In general, tabular methods need not overfit, and tuning parameters carefully can allow users to get relatively good performance while maintaining error rates between training and test.

Figure 3 left shows Equal Opportunity on the COMPAS dataset. To show that OxonFair can also work in low-data regimes where we have insufficient data for validation, we enforce fairness on the training set. As before, we binarize race to allow the use of AIF360. We drop race from the training data, and use inferred protected attributes to enforce fairness. Here OxonFair generates a frontier that is comparable or better than results from existing toolkits, and OxonFair+ (see Section A), further improves on these results. See Figure 3 right for a comparison with Fairlearn varying the groups.

### 5.1   Computer Vision and CelebA

**CelebA** [79]: We use the standard aligned & cropped partitions frequently used in fairness evaluation [17, 21, 28, 41]. Following Ramaswamy et al. [28], we consider the 26 *gender-independent*, *gender-dependent* and *inconsistently labelled* attributes as the target attributes for our evaluations (see Table 12 for details). *Male* is treated as the protected attribute.

**Implementation Details**   We follow Wang et al.'s setup [41]. We use a Resnet-50 backbone [80] trained on ImageNet [81]. A multitask classification model is trained, replacing the final fully-connected layer of the backbone with a separate fully-connected head that performs binary prediction for all attributes. Dropout [82] (p = 0.5) is applied. All models are trained with a batch size of

---

[6]This dataset is carefully curated and balanced. To induce unfairness we altered the sampling and dropped half the people recorded as male and that did not have medical complications across the entire dataset.

|              | ERM  | Uniconf. Adv [74] | Domain Disc. [41] | Domain Ind. [41] | OxonFair DEO |
|--------------|------|-------------------|-------------------|------------------|--------------|
| **Gender-Dependent Attributes** | | | | | |
| Acc. ($\uparrow$) | **86.7** | 86.1 | 86.6 | 85.6 | 85.8 |
| DEO ($\downarrow$) | 26.4 | 25.0 | 21.9 | 6.50 | **3.92** |
| **Inconsistently Labelled Attributes** | | | | | |
| Acc. ($\uparrow$) | 83.0 | 82.5 | **83.1** | 82.3 | 82.1 |
| DEO ($\downarrow$) | 21.9 | 29.1 | 25.3 | 17.2 | **2.36** |

Table 2: A comparison against standard vision approaches on the more challenging CelebA attributes. OxonFair continues to work well here. All methods share a common backbone and training process. An extended version of this table that considers minimax fairness can be found in Table 14.

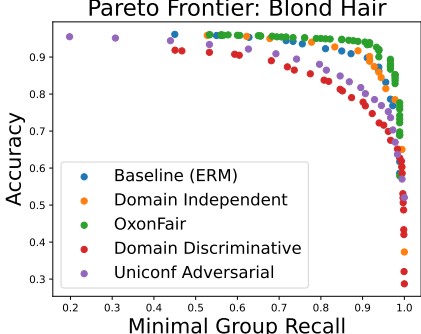

| CelebA | $\delta = 0.50$ | $\delta = 0.75$ | $\delta = 0.90$ |
|--------|-----------------|-----------------|-----------------|
| Baseline | 89.0 | 84.5 | 77.6 |
| Adversarial | 87.8 | 82.4 | 75.2 |
| Domain-Dep | 82.3 | 76.8 | 68.6 |
| Domain-Ind | 89.2 | 86.2 | 79.8 |
| OxonFair | **89.9** | **87.3** | **81.8** |

Figure 4: **Left:** The Pareto frontier of min. group recall vs. accuracy on `Blond Hair` demonstrates OxonFair's superior performance. **Right:** Comparing accuracy of fairness methods on 26 CelebA attributes while varying global decision thresholds to increase the minimum group recall level to $\delta$.

32 is and using Adam [83] (learning rate 1e-4). We train for 20 epochs and select the model with highest validation accuracy. Images are center-cropped and resized to $224 \times 224$. During training, we randomly crop and horizontally flip the images. See Appendix E.

**Results:** Table 1 and 2 demonstrates that using OxonFair as described in Section 4.2 generates fairer and more accurate solutions on unseen test data than other fair methods. Simple approaches such as Domain Independent training were more effective than adversarial training for enforcing fairness confirming [41, 43]. Occasionally, OxonFair finds solutions on the Pareto frontier that are both fairer and more accurate than the unconstrained classifier (See Figure 5).

Figure 4 shows a novel fairness evaluation motivated by medical use cases [8, 42] where practitioners might want to correctly identify at least $\delta\%$ of positive cases in each group. We evaluate how accuracy changes if we guarantee that the minimum recall is above $\delta\%$ for every group. For OxonFair, we call `.fit(accuracy, recall.min, ` $\delta$`)`. For other methods, we vary a global offset to ensure that the minimum recall is at least $\delta$.

## 5.2 NLP and Toxic Content

We conducted experiments on hate speech detection and toxicity classification using two datasets: the multilingual Twitter corpus [84] and Jigsaw [85]. Experiments were performed across five languages (English, Polish, Spanish, Portuguese, and Italian) and five demographic factors (age, country, gender, race/ethnicity, and religion) were treated as the protected groups. For details, see Appendix F.1.

We compare OxonFair with the following approaches. **Base** reports results of the standard BERT model [86]. **CDA** (Counterfactual Data Augmentation) [29, 87–90] rebalances a corpus by swapping bias attribute words (e.g., he/she) in a dataset based on a given dictionary. **DP** (Demographic Parity) uses regularization [23, 43] to enforce DP. **EO** (Equal Opportunity [3]) uses the regularization of [23, 43] to enforce EO. **Dropout** [88, 90] is used as a regularization technique [82] for bias mitigation and improving small group generalization. **Rebalance** [11, 91] method resamples the minor groups to the same sample size as other groups to mitigate bias. We report scores on Oxonfair optimized for

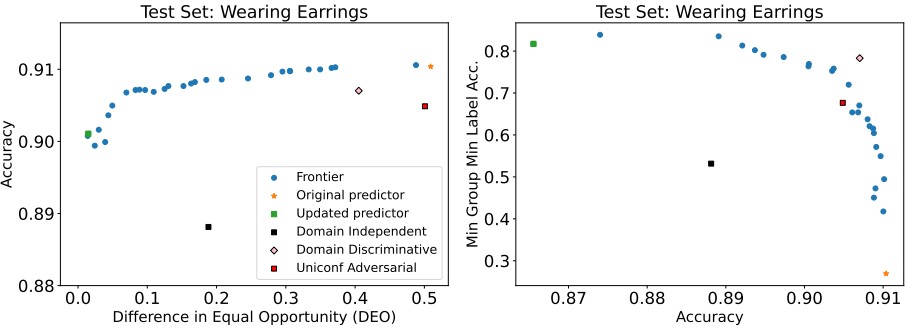

Figure 5: The Pareto frontier on test data when enforcing two fairness measures (DEO and Min Group Min Label Acc; see Appendix C.1) for the Earrings attribute. Inspecting the Pareto frontier shows a wide range of solutions, including some that improve fairness while retaining similar accuracy.

Table 3: Multilingual Twitter dataset: Gender (2 groups: Male, Female).

|  | F1 (↑) | Balanced Acc. (↑) | Acc. (↑) | DEO (↓) |
|---|---|---|---|---|
| Base | 40.8 | 63.2 | 89.8 | 21.4 |
| CDA [29] | 43.2 | 64.4 | 89.8 | 16.0 |
| DP [23] | 37.2 | 61.7 | 89.5 | 17.9 |
| EO [3] | 32.0 | 59.6 | 89.1 | 13.2 |
| Dropout [88] | 32.2 | 59.8 | 88.9 | 13.8 |
| Rebalance [11] | 38.2 | 62.1 | 89.5 | 19.1 |
| OxonFair Ac. | 34.1 | 60.7 | 88.5 | 8.45 |
| OxonFair F1 | 44.6 | 69.1 | 84.7 | 2.10 |
| OxonFair B. Ac. | 47.1 | 71.2 | 84.8 | 7.33 |
| OxonFair*[7] Ac. | 40.0 | 63.3 | 89.0 | 5.99 |
| OxonFair* F1 | 47.5 | 70.9 | 85.5 | 5.59 |
| OxonFair* B. Ac. | 40.8 | 64.6 | 87.3 | 13.0 |

Table 4: Jigsaw dataset: Religion (3 groups: Christian, Muslim, Other).

|  | F1 (↑) | Balanced Acc. (↑) | Acc. (↑) | DEO (↓) |
|---|---|---|---|---|
| Base | 42.1 | 74.8 | 75.0 | 7.33 |
| CDA [29] | 40.4 | 73.8 | 73.0 | 8.98 |
| DP [23] | 44.5 | 69.2 | 85.5 | 3.68 |
| EO [3] | 41.1 | 68.8 | 82.2 | 4.60 |
| Dropout [88] | 42.7 | 74.1 | 77.0 | 7.94 |
| Rebalance [11] | 39.1 | 73.7 | 70.3 | 9.67 |
| OxonFair Ac. | 33.7 | 60.5 | 89.2 | 2.36 |
| OxonFair F1 | 44.4 | 69.5 | 85.0 | 3.79 |
| OxonFair B. Ac. | 42.2 | 74.2 | 76.1 | 4.78 |
| OxonFair* Ac. | 26.3 | 57.6 | 88.8 | 1.84 |
| OxonFair* F1 | 44.3 | 68.5 | 86.2 | 0.84 |
| OxonFair* B. Ac. | 41.9 | 73.7 | 76.3 | 4.56 |

different metrics: Accuracy; F1 Score; and Balanced Accuracy, and always enforce that the DEO is under 5% on the validation set.

**Results:** Results are shown in Tables 3 and 4. Our observations indicate that: 1) all debiasing methods improve the equal opportunity score and help mitigate bias on Twitter, but not on jigsaw. 2) our toolkit consistently reduces the difference in equal opportunity more than any other approach; 3) for 4/6 experiments we actually improve the objective over the baseline while enforcing fairness, showing the value in targeting a particular objective. For additional experiments on multilingual and multi-demographic data, and the Jigsaw race data, see Appendix F.2, and Appendix F.3.

While improving fairness more than existing approaches, OxonFair performs substantially worse on these NLP datasets than on CelbA. There are two challenges present in these datasets but not in CelbA: *(i)* it is likely much harder to infer gender or religion from short text data than it is to infer gender from a photo of a face. *(ii)* The limited number of positively labelled datapoints examples that makes estimating DEO, F1,and balanced accuracy unstable (see Tables 16 and 17 for details). To better understand the influence of the two factors we refit OxonFair using the true rather than inferred attributes at test time (bottom block) and see no reliable improvements, suggesting that we are most limited by data scarcity.

---

[7]Here OxonFair indicates the use of inferred group membership, while OxonFair* uses true group membership at test time.

# 6 Conclusion

The key contributions of our toolkit lie in being more expressive than other approaches, and supporting NLP and computer vision. Despite this, most of the experiments focus on the standard definitions of Demographic Parity and Equal Opportunity. This is not because we agree that they are the right measures, but because we believe that the best way to show that OxonFair works is to compete with other methods in what they do best. On low-dimensional tabular data, when optimizing accuracy and a standard fairness measure, it is largely comparable with Fairlearn, but if overfitting or non-standard performance criteria and fairness metrics are a concern, then OxonFair has obvious advantages. For NLP, and computer vision, our approach clearly improves on existing state-of-the-art. In no small part, this is due to the observation of [17], that methods for estimating or enforcing error-based fairness metrics on high-capacity models that do not use held-out validation data can not work.

We hope that OxonFair will free policy-makers and domain experts to directly specify fairness measures and objectives that are a better match for the harms that they face. In particular, we want to call out the measures in Figure 8 as relevant to medical ML. The question of how much accuracy can we retain, while guaranteeing that classifier sensitivity (AKA recall) is above k% for every group, captures notions of fairness and clinical relevance in a way that standard fairness notions do not [8].

**Limitations:** We have chosen to optimize as broad a set of formulations as possible. As a result, for certain metrics (particularly equalized odds [3]) the solutions found are known to be suboptimal[8]; and for others [12] the exponential search is unneeded. Techniques targeting particular formulations may be needed to address this.

A key challenge for most fairness approaches is in obtaining the group labels used to measure unfairnesss, and we are no exception. In particular, the gender labels in CelebA and the race and religion labels in our NLP experiments consist of a small number of externally assigned labels that may not match how people self-identify. Improving and measuring fairness with respect to these coarse labels can miss other forms of inequality. Moreover, a major driver of unfairness is a lack of data regarding particular groups. However, this very absence of data makes it hard for any toolkit to detect or rectify unfairness.

**Broader Impact:** OxonFair is a tool for altering the decisions made by ML systems that are frequently trained on biased data. Care must be taken that fair ML is used as a final step after correcting for bias and errors in data collation, and not as a sticking plaster to mask problems [92]. Indeed, inappropriate uses of fairness can lock in biases present in training [72]. Under the hood, OxonFair performs a form of positive discrimination, where we alter scores in response to (perceived) protected characteristics to rectify specific existing inequalities[9]. As such, there are many scenarios where its use may be inappropriate for legal or ethical reasons.

# 7 Acknowledgements

This work has been supported through research funding provided by the Wellcome Trust (grant nr 223765/Z/21/Z), Sloan Foundation (grant nr G-2021-16779), Department of Health and Social Care, EPSRC (grant nr EP/Y019393/1), and Luminate Group. Their funding supports the Trustworthiness Auditing for AI project and Governance of Emerging Technologies research programme at the Oxford Internet Institute, University of Oxford.

An early prototype version of the same toolkit, for tabular data, was developed while CR was working at AWS and is available online as autogluon.fair (`https://github.com/autogluon/autogluon-fair/`). CR is grateful to Nick Erickson and Weisu Yin for code reviews of the prototype. The authors thank Kaivalya Rawal for feedback on the manuscript and testing the codebase.

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

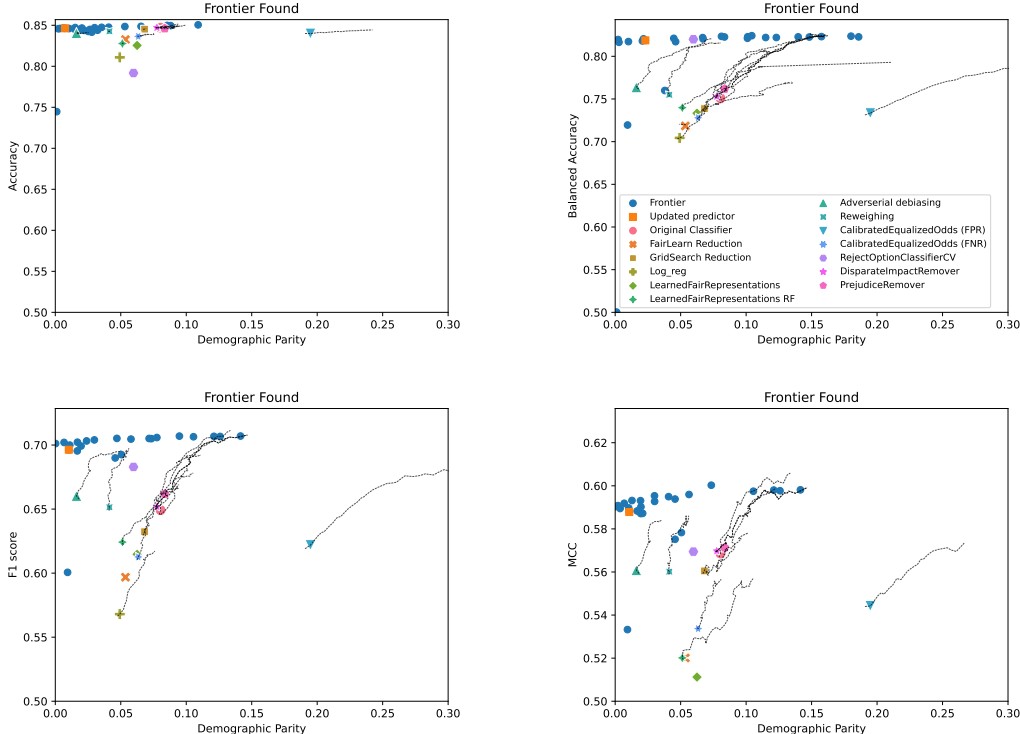

Figure 6: We show a full comparison of the methods provided by AIF360 and Fairlearn on the adult dataset with 4 different choices of metric (accuracy, balanced accuracy, F1, and Matthews correlation coefficient (MCC)), while enforcing demographic parity. We follow the design decisions of [9] and use a random forest with 100 trees and a minimum leaf size of 20. Only OxonFair allows the specification of an objective, and for all other methods we try to alter the decision threshold to better optimize the objective. However, as we improve the objective, we see fairness deteriorates. To avoid this, OxonFair jointly optimize both a fairness measure and an objective.

## A  Inferred characteristics

In many situations, protected attributes are not available at test time. In this case, we simply use inferred characteristics to assign per-group thresholds and adjust these thresholds to guarantee fairness with respect to the true (i.e. uninferred) groups.

When using inferred characteristics, we offer two pathways for handling estimated group membership. The first pathway we consider makes a hard assignment of individuals to groups, based on a classifier response. The second pathway explicitly uses the classifier confidence as part of a per-datapoint threshold. In practice, we find little difference between the two approaches, but the hard assignment to groups is substantially more efficient and therefore allows for a finer grid search and generally better performance. However, the soft assignment remains useful for the integration of our method with neural networks, where we explicitly merge two heads of a neural network to arrive at a single fair model.

### A.1  Fast pathway

The fast pathway closely follows the efficient grid search for known characteristics. We partition the dataset by inferred characteristics, and then repeat the trick. However, as the inferred characteristics do not need to perfectly align with the true characteristics, we also keep track of the true group datapoints belongs to, i.e., for all datapoints assigned to a particular inferred group, we compute the cumulative sum of positives and negatives that truly belong to each group. This allows us to vary the thresholds with respect to inferred groups while computing group measures with respect to the

true groups. This can be understood as replacing the decision function (1) with $f(x) - t \cdot G'(x) \geq 0$ where $G'$ is a binary vector valued function that sums to 1, but need not correspond to $G$ exactly.

This explicit decoupling of inferred groups from the true group membership allows us to consider partitionings of the data that do not align with group membership. We found it particularly helpful to include an additional 'don't know' group. By default, any datapoint assigned a score[10] from the classifier below $2/3$ is assigned to this group, and receives a different threshold to those datapoints that the classifier is confident about. The improved frontiers are shown in the tabular experimental section as OxonFair+, where they offer a clear advantage over our baseline OxonFair.

### A.2   Slow pathway

The slow pathway tunes $t$ to optimize the decision process $f(x) - t \cdot g(x) \geq 0$, where $g$ is a real vector valued function. Given the lack of assumptions, no obvious speed-up was possible and we perform a two stage naïve grid-search, first coarsely to extract an approximate Pareto frontier, and then a finer search over the range of thresholds found in the first stage. This is then followed by a final interpolation that checks for candidates around pairs of adjacent candidates currently in the frontier.

In situations where $g(x)$ is the output of a classifier and $G'(x)$ its binarization, it is reasonable to suspect that loss of information from binarization might lead to a drop in performance when we compare the slow pathway with the fast. In practice, we never found a significant change, and in a like-with-like comparison over a similar number of thresholds the fast pathway was as likely to be fractionally better as it was to be worse. Moreover, for more than 3 groups the slow pathway becomes punitively slow, and to keep the runtime acceptable requires decreasing the grid size in a way that harms performance.

Despite this, we kept the slow pathway as it is directly applicable to deep networks as we describe in the next section. In practice, when working with deep networks we make use of a hybrid approach, and perform the fast and slow grid searches before fusing them into a single frontier and then performing interpolation. This allows us to benefit from the better solutions found by a fine grid search when the output of the second head is near binary (see Figure 2), and robustly carry over to the slower pathway where its binarization is a bad approximation of the network output.

## B   Implementation of Performance and Fairness Measures

To make OxonFair readily extensible, we create a custom class to implement all performance and fairness measures. As such, should OxonFair not support a particular measure, both the objectives and constraints can be readily extended by the end user.

Measures used by OxonFair are defined as instances of a python class `GroupMetrics`. Each group measure is specified by a function that takes the number of True Positives, False Positives, False Negatives, and True Negatives and returns a score; A string specifying the name of the measure; and optionally a bool indicating if greater values are better than smaller ones.

For example, accuracy is defined as:

```
accuracy = gm.GroupMetric(lambda TP, FP, FN, TN: (TP + TN) / (TP + FP + FN
+ TN), 'Accuracy')
```

For efficiency, our approach relies on broadcast semantics and all operations in the function must be applicable to numpy arrays. Having defined a GroupMetric it can be called in two ways. Either:

```
accuracy(target_labels, predictions, groups)
```

Here `target_labels` and `predictions` are binary vectors corresponding to either the target ground-truth values, or the predictions made by a classifier, with 1 representing the positive label and 0 otherwise. `groups` is simply a vector of values where each unique value is assumed to correspond to a distinct group.

The other way it can be called is by passing it a single 3D array of dimension 4 by number of groups by k, where k is the number of candidate classifiers that the measure should be computed over.

---

[10]User controllable threshold.

Table 5: The fairness measures in the review of [61]. All 9 group metrics that concern the decisions made by a classifier are supported by OxonFair.

| Vema and Rubin [61] Metrics | OxonFair name | Fairlearn |
|---|---|---|
| Group fairness or statistical parity | `demographic_parity` | Yes |
| Conditional statistical parity | `conditional_group_metrics.`
`    pos_pred_rate.diff` | No |
| Predictive parity | `predictive_parity` | No |
| False positive error rate balance | `false_pos_rate.diff` | Yes |
| False negative error rate balance | `false_neg_rate.diff` | Yes |
| Equalized odds | `equalized_odds` | Yes |
| Conditional use accuracy equality | `cond_use_accuracy` | No |
| Overall accuracy equality | `accuracy.diff` | No |
| Treatment equality | `treatment.diff` | No |
| Test-fairness or calibration | Not decision based | |
| Well calibration | Not decision based | |
| Balance for positive class | Not decision based | |
| Balance for negative class | Not decision based | |
| Causal discrimination | Individual fairness | |
| Fairness through unawareness | Individual fairness | |
| Fairness through awareness | Individual fairness | |
| No unresolved discrimination | Individual fairness | |
| No proxy discrimination | Individual fairness | |
| Fair inference | Individual fairness | |

As a convenience, GroupMetrics automatically implements a range of functionality as sub-objects.

Having defined a metric as above, we have a range of different objects:

- `metric.diff` reports the average absolute difference of the method between groups.

- `metric.average` reports the average of the method taken over all groups.

- `metric.max_diff` reports the maximum difference of the method between any pair of groups.

- `metric.max` reports the maximum value for any group.

- `metric.min` reports the minimum value for any group.

- `metric.overall` reports the overall value for all groups combined, and is the same as calling metric directly

- `metric.ratio` reports the average over distinct pairs of groups of the smallest value divided by the largest

- `metric.per_group` reports the value for every group.

All of these can be passed directly to fit, or to the evaluation functions we provide.

The vast majority of fairness metrics are implemented as a `.diff` of a standard performance measure, and by placing a `.min` after any measure such as recall or precision it is possible to add constraints that enforce that the precision or recall is above a particular value for every group.

**Total Metrics** Computing certain metrics, particular Conditional Demographic Disparity [68], and Bias Amplification [70], requires knowledge of the total number of TP etc. alongside the number of TP in each group.

To implement these measures, we support lambda functions that take per-group values, followed by the global values, giving 8 arguments in total. These lambda functions should be passed to `GroupMetric` in the same way along ith the optional argument `total_metric=True`.

Table 6: The post-training fairness measures in the review of [93]. All measures are supported by OxonFair.

| Post-training Metrics [93] | OxonFair name | Fairlearn |
|---|---|---|
| Diff. in pos. proportions in predicted labels | `demographic_parity` | Yes |
| Disparate Impact | `disparate_impact` | No |
| Difference in Conditional Acceptance | `cond_accept.diff` | No |
| Difference in Conditional Rejection | `cond_reject.diff` | No |
| Accuracy Difference | `accuracy.diff` | No |
| Recall Difference | `recall.diff` | Yes |
| Difference In Acceptance Rates | `acceptance_rate.diff` | No |
| Difference in Rejection Rates | `rejection_rate.diff` | No |
| Treatment Equality | `treatment_equality` | No |
| Conditional Demographic Disparity | `conditional_group_metrics.`
`pos_pred_rate.diff` | No |

Table 7: Enforcing fairness for all definitions in [93] on COMPAS with inferred attributes. We enforce the fairness definitions with respect to three racial groups, African American, Caucasian, and Other – consisting of all other labelled ethnicities. There are a total 350 individuals labelled 'Other' in the test set, making most metrics of fairness unstable and difficult to enforce. Nonetheless, we improve on all metrics. For all metrics except disparate impact, we enforce that the score on train is below 2.5% and for disparate impact we enforce that the score on train is above 97.5%. XGBoost is used as the base classifier, and the dataset is split into 70% train and 30% test.

| | Measure (original) | Measure (updated) | Accuracy (original) | Accuracy (updated) |
|---|---|---|---|---|
| Demographic Parity | 0.148706 | 0.097142 | 0.661345 | 0.620588 |
| Disparate Impact | 0.668305 | 0.740940 | 0.661345 | 0.605042 |
| Difference in Conditional Acceptance Rate | 0.231862 | 0.151159 | 0.661345 | 0.642857 |
| Difference in Conditional Rejectance Rate | 0.048625 | 0.025138 | 0.661345 | 0.655882 |
| Difference in Accuracy | 0.013172 | 0.006351 | 0.661345 | 0.665546 |
| Difference in Recall | 0.151210 | 0.105154 | 0.661345 | 0.612185 |
| Difference in Acceptance Rate | 0.070072 | 0.066591 | 0.661345 | 0.662605 |
| Difference in Specificity | 0.097490 | 0.064139 | 0.661345 | 0.660504 |
| Difference in Rejection Rate | 0.050085 | 0.050215 | 0.661345 | 0.661345 |
| Treatment Equality | 0.201717 | 0.105115 | 0.661345 | 0.660924 |
| Conditional Demographic Parity | 0.150927 | 0.073203 | 0.661345 | 0.626471 |

## C  Additional Metrics

To demonstrate OxonFair's versatility, Tables 5 and 6 show the metrics of two review papers and how many can are implemented out of the box by our approach. An example showing how all clarify metrics can be enforced using inferred groups, and three group labels on compas can be seen in Table 7.

### C.1  Minimax Fairness

Minimax fairness [59, 66, 67] refers to the family of methods which minimize the loss of the group where the algorithm performs worst, i.e., they minimize the maximal loss. [94] observed that sufficiently expressive classifiers, such as those considered by this paper, including boosting, random forests, or deep networks on image and NLP tended to be per group optimal, when the groups do not correspond to the predicted label. As such they are already minimax optimal and the solutions found by minimax fairness methods are indistinguishable from those found by empiric risk minimization. This still leaves the case where groups include the label (for example, the groups may correspond to the product of gender and the variable we are trying to predict, such as sick or not sick). In this case, as convincingly shown by [59], the solutions found do not correspond to ERM.

Here, we compare OxonFair against minimax fairness. To do this, we define a new performance measure corresponding to the lowest accuracy over the positive or negative labelled datapoints.

$$\text{min accuracy} = \min \left( \frac{TP}{TP + FP}, \frac{TN}{FN + TN} \right) \tag{5}$$

Martinez et al [59] argued that we should seek a Pareto optimal solution that has the highest possible overall accuracy, subject to the requirement it maximizes the lowest per group accuracy. We can do this in OxonFair by calling `fpredictor.fit(gm.min_accuracy.min, gm.accuracy, 0)` Here `min_accuracy.min` corresponds to the lowest min accuracy of any group. We use accuracy $> 0$ as the constraint, as we do not want an active constraint from preventing us from finding the element of the Pareto frontier (see [59] for frontier details) with the highest minimum accuracy. Note that the groups used by OxonFair with this loss correspond to the true groups, such as ethnicity or gender, while the groups used by minimax fairness are the product of these groups with the target labels. Existing methods for minimax fairness optimize the same loss and have indistinguishable accuracy, only differing in the speed of convergence[67] As such, in Table 8 we only report results for a variant of [67].

Similarly, in computer vision [17], optimized the same objective by iteratively generating synthetic data for the worst performing group, where groups were defined as the product of ground-truth labels, and sex. We compare against them in Table 15.

| XGBoost: Adult (sex) | Min Accuracy | Overall Accuracy |
|---|---|---|
| ERM Training | 70.3% | 90.9% |
| Minimax Training [67]. | 85.2% | 88.9% |
| ERM Validation | 58.8% | 86.9% |
| Minimax Validation [67]. | 76.2% | 83.9% |
| OxonFair Validation | 79.1% | 84.4% |
| ERM Test | 59.6% | 86.6% |
| Minimax Test [67]. | 77.9% | 84.1% |
| OxonFair Test | 80.5% | 84.6% |

Table 8: Results for XGBoost: Adult (sex)

## C.2   Utility Optimization

OxonFair supports the utility-based approach of Bakalar et al. [60], whereby different thresholds can be selected per group to optimize a utility-based objective. Utility functions can be defined in one line. In the following example, we consider a scenario where an ML system identifies issues that may require interventions. In this example, every intervention has a cost of 1, regardless of if it was needed, but a missed intervention that was needed has a cost of 5. Finally, not making an intervention when one was not needed has a cost of 0.

Figure 7 shows code where `fpredictor` minimizes the utility subject to the requirement that the overall recall cannot drop below 0.5.

Figure 7: Defining and optimizing a custom utility function with OxonFair [60].

## C.3   Levelling up

One criticism of many methods of algorithmic fairness is that enforcing equality of recall rates (as in equal opportunity) or selection rates (as in demographic parity) will decrease the recall/selection rate for some groups while increasing it for others. This behavior is an artifact of trying to maximize accuracy [8] and occurs despite fairness methods altering the overall selection rate [95]. As an alternative, OxonFair supports **levelling up** where harms are reduced to, at most, a given level per group [8]. For example, if we believe that black patients are being disproportionately harmed by a high number of false negatives in cancer detection (i.e., low recall), instead of enforcing that these properties be equalized across groups, we can instead require that every group of patients has, at

least, a minimum recall score. Depending on the use case, similar constraints can be imposed in with respect to per-group minimal selection rates, or minimal precision. These constraints can be enforced by a single call, for example, enforcing that the precision is above 70% while otherwise maximizing accuracy can be enforced by calling: `.fit(gm.accuracy, gm.precision.min, 0.7)`. See also Figure 4.

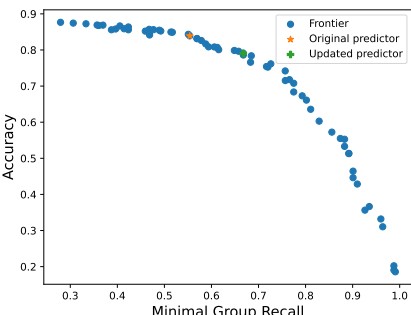

Figure 8: Levelling-up with OxonFair by imposing a minimum group recall of 0.7 on the Fitzpatrick-17k [96] validation set - `fpredictor.fit(gm.accuracy, gm.recall.min, 0.7)`.

To enforce levelling up as a hard additional constraint, `fit` takes an optional argument, `force_levelling_up`. Setting this equal to `'+'` forces the selection rate to increase in the search procedure, while setting it equal to `'-'` means it can only decrease. As most performance metrics underlying fairness constraints (e.g. recall, selection rate, precision) are monotonic with respect to the selection rate [95], this can prevent levelling down.

The levelling-up constraint can be combined with standard forms of fairness e.g. by calling `.fit(gm.accuracy, gm.demographic_parity, 0.01, force_levelling_up='+')` but they are most useful when using levelling up constraints in conjuncture with an inadequately optimized classifier. In some circumstances calling `.fit(gm.accuracy, gm.recall.min, 0.6)` can result in a decrease in recall rate for some groups providing this increases accuracy and does not drop below the recall below the specified minimal rate.

## C.4 Fairness under constrained capacity

When deploying fairness in practice, we may be capacity limited. For example, as in Figure 8 we may use the output of a classifier for detecting cancer to schedule follow-up appointments. In such a case, you might wish that the recall is high for each demographic group, but be constrained by the number of available appointments. Calling `.fit(gm.recall.min, gm.pos_pred_rate, 0.4, greater_is_better_const=False)` will maximize the recall on the worst-off group subject to a requirement that no more than 40% of cases are scheduled follow-up appointments.

In general, maximizing the group minimum of any measure that is monotone with respect to the selection rate, while enforcing a hard limit on the selection rate will enforce equality with respect to that measure (e.g. optimizing `gm.recall.min` will result in equal recall a.k.a. equal opportunity, while maximizing `gm.pos_pred_rate.min` will result in demographic parity), while also enforcing the selection rate constraints. See [95] for proof and a discussion of the issues arising, and [97] for an alternate approach.

As such, calling `.fit(gm.recall.min, gm.pos_pred_rate, k, greater_is_better_const = False)` will enforce equal opportunity at $k\%$ selection rate, and `.fit(gm.pos_pred_rate.min, gm.pos_pred_rate, 0.4, greater_is_better_const = False)` will enforce demographic parity at $k\%$ selection rate.

## C.5 Conditional Metrics

A key challenge of using fairness in practice is that often some sources of bias are known, and the practitioner is expected to determine if additional biases exist and to correct for them. For example, someone's salary affects which loans they are eligible for, but salary has a distinctly

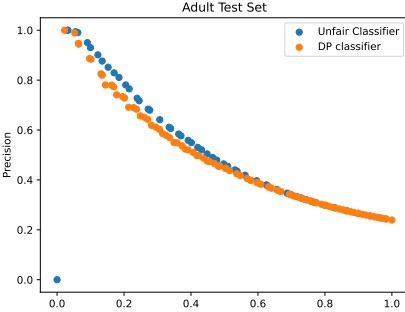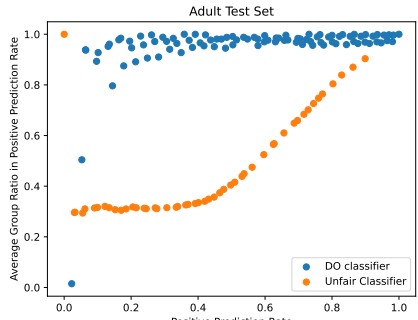

Figure 9: Solutions found when enforcing demographic parity with varying rate constraints. See Appendix C.4. **Left:** the change in precision as we enforce demographic parity. Note that we report precision as it is more informative than accuracy for low selection rates. **Right:** The ratio between selection rates (i.e. disparate impact) for different groups. We report the ratio rather than the difference, as the difference tends to zero as the selection rate also tends to zero. However, as the right figure shows, this ratio becomes unstable as the rate tends to zero.

different distribution for different ethnicities and genders. [65]. Identifying and correcting fairness here rapidly becomes challenging, when considering the intersection of attributes, many small groups arise and purely by chance some form of unfairness may be observed [98, 68]suggested the use of a technique from descriptive statistics that [99] had previously applied to the problem of schools admissions at Berkley [100]. In this famous example, every school in Berkley showed little gender bias, but due to different genders applying at different rates to different schools, and the schools themselves having substantially different acceptance rate, a strong overall gender bias was apparent.

[99] observed that you could correct for this bias by computing the per school selection-rate, and then taking a weighted average, where the weights are given by the total number of people applying to the school. The resulting selection rates are equivalent to a weighted selection-rate over the whole population, where the weight $w_i$ for an individual $i$ in a particular group and applying to a particular school is $w_i = \frac{\#\text{individuals in school}}{\#\text{individuals in group and school}}$. To enforce this form of conditional demographic parity in OxonFair, we simply replace the sum of true positives etc. in Section 3, with the weighted sum. We support a range of related fairness metrics, including conditional difference in accuracy; and conditional equal opportunity (note that for equal opportunity we replace the numbers used to compute $w_i$ with the same counts but only taking into account those that have positive ground-truth labels). As such metrics can level down (Appendix C.3), we also support conditional minimum selection rates, and conditional minimum recall.

In addition to this, we support the conditional fairness metrics based on EU/UK law of [68]. These measure the (weighted) proportion of people belonging to a particular group in the set of advantaged or disadvantaged people. These metrics can be computed for both the input target variables a classifier is trained to predict and for the classifier outputs.

### C.6    Bias Amplification Metrics

We also support variants of Bias Amplification, as defined by Zhao et al. and Wang et al. [69, 70].

As with the other metrics, we focus on scenarios where ground-truth group assignments exist even if they are unavailable at test time, and as such we focus on the Attribute → Task Bias (see [70]).

Bias Amplification is implemented as a per-group metric, `gm.bias_amplification`. As this measure is signed (a negative measure indicates the classifier reverses the bias present in the dataset), directly optimising it results in classifiers strongly biased in a new direction. Instead, we minimize the per group absolute bias amplification. The derivation is given below.

Following the notation of Wang et al. [70], let $\mathcal{A}$ be the set of protected demographic groups: for example, $\mathcal{A} = \{\text{male, female}\}$. $A_a$ for $a \in \mathcal{A}$ is the binary random variable corresponding to the presence of the group $a$; thus $P(A_{\text{woman}} = 1)$ can be empirically estimated as the fraction of images

in the dataset containing women. Let $T_t$ with $t \in \mathcal{T}$ similarly correspond to binary target tasks. Let $\hat{A}_a$ and $\hat{T}_t$ denote model predictions for the protected group $a$ and the target task $t$, respectively.

$$\text{BiasAmp}_{\rightarrow} = \frac{1}{|\mathcal{A}||\mathcal{T}|} \sum_{a \in \mathcal{A}, t \in \mathcal{T}} y_{at} \Delta_{at} + (1 - y_{at})(-\Delta_{at})$$

$$y_{at} = \mathbb{1}\left[ P(A_a = 1, T_t = 1) > P(A_a = 1)P(T_t = 1) \right]$$

$$\Delta_{at} = \begin{cases} P(\hat{T}_t = 1|A_a = 1) - P(T_t = 1|A_a = 1) \\ \text{if measuring } Attribute \rightarrow Task\ Bias \\ P(\hat{A}_a = 1|T_t = 1) - P(A_a = 1|T_t = 1) \\ \text{if measuring } Task \rightarrow Attribute\ Bias \end{cases} \quad (6)$$

Of which, the Attribute $\rightarrow$ Task Bias is relevant here.

Each component can be written as a function of the global True Positives, False Positives etc., and the per group True Positives, and as such it can be optimized by our framework, albeit, not by using a standard group metrics. However, this metric is gamable, and consistently underestimating labels in groups where they are overrepresented and vice versa would be optimal, but undesirable behavior that leads to a negative score.

Instead, we consider the absolute BiasAmp:

$$|\text{BiasAmp}|_{\rightarrow} = \frac{1}{|\mathcal{A}||\mathcal{T}|} \sum_{a \in \mathcal{A}, t \in \mathcal{T}} |y_{at} \Delta_{at} + (1 - y_{at})(-\Delta_{at})|$$

$$= \frac{1}{|\mathcal{A}||\mathcal{T}|} \sum_{a \in \mathcal{A}, t \in \mathcal{T}} |\Delta_{at}| \quad (7)$$

We can decompose $|\Delta_{at}|$ into the appropriate form for a `GroupMetric` (see Appendix B) as follows:

$$\Delta_{at} = P(\hat{T}_t = 1|A_a = 1) - P(T_t = 1|A_a = 1) \quad (8)$$

$$\Delta_{at} = \frac{TP + FN}{TP + TN + FP + FN} - \frac{TP + FP}{TP + TN + FP + FN} \quad (9)$$

$$|\Delta_{at}| = \left| \frac{FN - FP}{TP + TN + FP + FN} \right| \quad (10)$$

This will give a per group estimate of the absolute bias amplification, and calling its `.average` method will give the absolute bias amplification over all groups.

## D  Comparisons with specialist methods

### D.1  Fairret

Fairret [56] is a recently published toolkit for enforcing fairness in pytorch networks, using standard fairness constraints [23]. This toolkit has only been shown to work for tabular data. There are mathematical reasons to think that it will not work for *bias preserving* fairness metrics [72], such as equalized opportunity, on computer vision or NLP classifiers where classifiers obtain zero error on the training set [17], and therefore trivially satisfy all *bias preserving* fairness metrics. However, as Fairret uses standard relaxations, it should have comparable accuracy/fairness trade-offs to OxonFair when weakly enforcing demographic parity on image data (see [21]).

Fairret should also have comparable performance on NLP data to the DP and EO regularized approaches reported in the main body of the paper.

Here we focus on using Fairret to enforce Equal Opportunity on tabular data as shown in their paper. Figure 10 shows a comparison with Oxonfair. For a like-with-like comparison between Fairret and OxonFair we use 70% of the data for training for Fairret (which requires no validation data), and for OxonFair split this into 40% training data and 30% validation data.

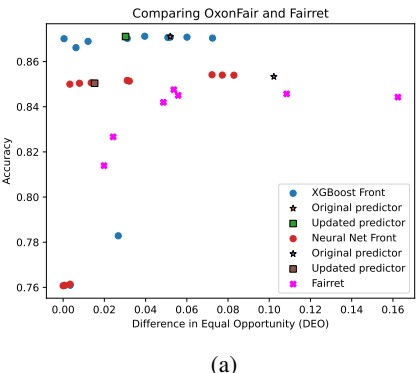
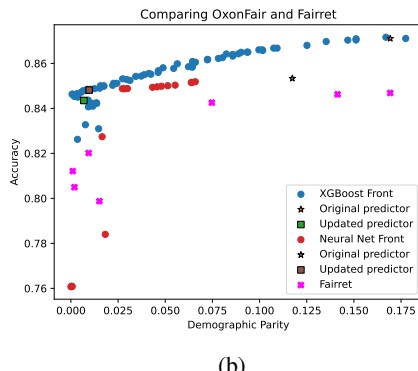

| (a) | (b) |

Figure 10: Comparing OxonFair and Fairret [56] on adult using sex as the protected attribute. A simple neural network classifier with two hidden layers is used as the base classifier. Fairness strengths are varied for different Fairret implementations. Difference in Equal Opportunity and Demographic Parity are considered. An OxonFair-based frontier using XGBoost is also displayed.

In general, neural networks perform worse than boosting on tabular data, and should not be used where maximizing accuracy is a concern. Nonetheless, we see that Fairret shows worse accuracy/fairness trade-offs than OxonFair on neural networks with the same architecture.

Compared to OxonFair, there are three challenges faced by Fairret that might be contributing to its worse performance.

- The enforced constraints are a relaxation of the underlying integer fairness constraints, and even when completely satisfied, they need not imply fairness [30].

- A mismatch between errors on the training and test set. Even when models do not obtain zero training error they can still overfit, and minimizing equal opportunity on the training set does not imply that it is optimized on the test set.

- Failure to converge. To induce stability in the performance of Fairret we had to use a much larger minibatch size of 1000, as the minibatch must be large enough to estimate the EO violation somewhat stably if we want the method to converge. It is possible that either the batch size was still not large enough or that it was so large that it caused issues with optimizing the log loss.

However, other issues might be the reason for the performance discrepancy.

### D.2 Specialist Equalized Odds Solvers

In this section, we show how OxonFair can be adapted through the use of custom groups to mimic the performance of Specialist Equalized Odds solvers. We compare against the recently published [57], which was shown to outperform a wide range of existing methods. Like us, [57] assigns thresholds on a per-group basis to enforce fairness, and enforces fairness up to a prespecified threshold (e.g., equalized odds is less than 5%) while maximizing accuracy. Unlike the default behaviour of Oxonfair, however, it randomly assigns members of each group one of two different thresholds. To mimic this behaviour, we make use of the same trick used in Section A.1, namely that the 'inferred groups' used to assign thresholds are not expected to align with the true groups, and that we can introduce additional groups to increase the expressiveness of the model.

Before showing how to enforce Equalized Odds, we strongly recommend that this is not used in practice. [57] also notes that enforcing it is contentious. Thresholding methods such as OxonFair and [57] can be understood as methods that trade-off sensitivity or recall against specificity. When enforcing Equalized odds, we will move to a new point on the sensitivity specificity curve, for the worst performing group, while degrading the performance for all other groups. Viewed through the lens of levelling up [8], and which specific harms are incurred by each group, versus a base classifier: enforcing equalized odds will increase one of the sensitivity and specificity for the worst performing group and decrease the other; and may decrease sensitivity and specificity for all other

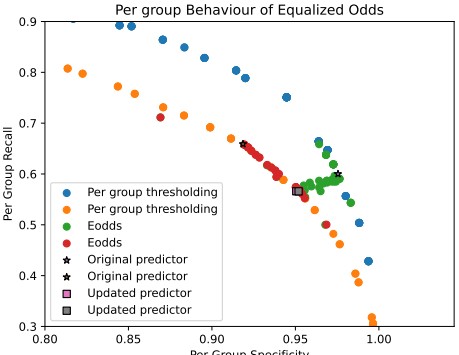
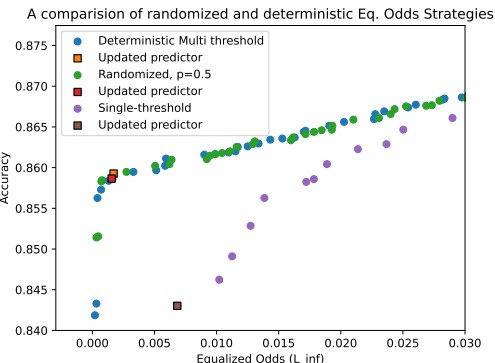

Figure 11: **Left:** A close-up of possible per-group trade-offs when enforcing Equalized odds. This figure shows possible behaviour when enforcing Equalized odds with respect to sex on the adult dataset. Compared to the original predictor, we see a substantial decrease in recall for the worst performing group accompanied by a small increase in specificity. For the better performing group, both recall and specificity are decreased in order to enforce fairness. **Right:** The accuracy/fairness trade-offs of OxonFair using single threshold, deterministic multi-thresholds, and randomized multi-thresholds. Single threshold performs substantially worse for strong fairness constraints, but the other two strategies are interchangeable. All results shown are on validation data.

groups. (see Figure Figure 11 for an illustration of the frontier found by OxonFair with respect to recall and specificity). While other methods are harder to analyze than thresholding, the fact that they have worse accuracy/fairness trade-offs suggests that they deteriorate classifiers more than simple thresholding.

We consider four threshold sets for OxonFair.

1. Default OxonFair using one threshold per group.

2. A randomized version that mimics the behaviour of [57], by assigning members of each group to two subgroups with 50/50 probability.

3. As randomized approaches can be criticized for their instability, we also consider a deterministic version that flips the scores of positively labelled data points scored below a threshold, and also flips the score of negatively labelled points scored above a threshold.

4. An inferred variant that does the same as 3, but doesn't require access to groups at test time.

Additional code for the last three options is shown below. These functions can be passed to `FairPredictor` using the `inferred_groups` option.

Our comparison with [57] can be summarized as the randomized and deterministic methods are broadly interchangeable both with [57] (see Figure 12) and with each other (see Figure 11 right), and they strongly outperform the default single-threshold OxonFair for strong fairness constraints.

As both [57] and multi-threshold OxonFair solve the same optimization problem via different routes (a specialist LP solver for [57], grid-search for OxonFair) we expect them to obtain similar solutions, and this we find in Figure 12. However, [57] makes use of a formulation that is more efficient for large numbers of thresholds, but less expressive and harder to adapt to infered group membership, or new fairness or performance constraints. As such, it is unsurprising that it is substantially more efficient than grid-search over the 8 thresholds used in these experiments – at least when finding a solution with e.g. an Eodds violation of no more than 0.05. What was more surprising was that when using the code of [57] to compute an entire fairness/accuracy Pareto frontier, Oxonfair remained more efficient. See Table 9 for details. This remains a reminder that big-O notation is uninformative with respect to runtime, when our datasets are extremely limited in their number of groups.

| Method | [57] (single fairness eval) | [57] (Complete Frontier) | OxonFair Complete Frontier |
|---|---|---|---|
| Runtime | 3 sec | 4m49.1 sec | 1m18s |

Table 9: Runtime comparison between [57] and OxonFair, on the Folktables dataset [101], using four racial groups. This represents the largest problem with the most groups reported in [57], and as such is the experiment where we would expect [57] to outperform OxonFair the most with respect to runtime. While [57] is faster if enforcing fairness to a known amount, e.g., maximizing accuracy subject to EOdds<0.05%, OxonFair remains faster for computing the entire frontier.

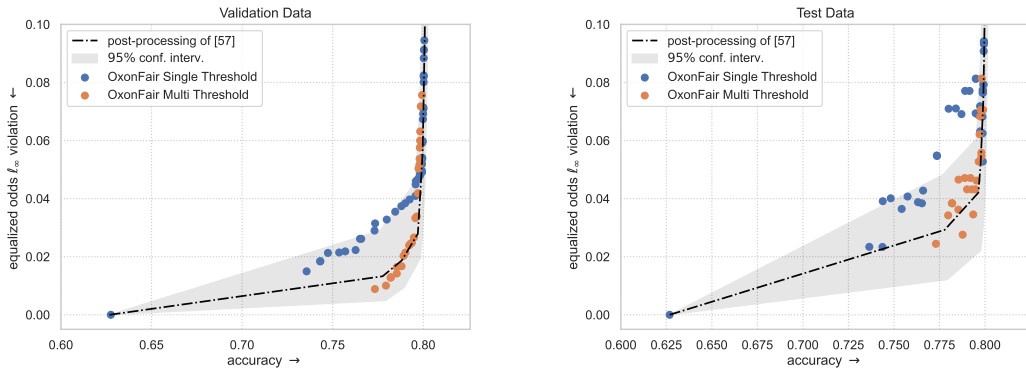

Figure 12: A comparison between single-threshold OxonFair and deterministic multi-threshold OxonFair with [57]. As expected, Multi-threshold is directly comparable to [57], with single threshold performing worse.

| | Original | OxonFair Multi | OxonFair + | FairLearn |
|---|---|---|---|---|
| Accuracy | 0.871016 | 0.866921 | 0.859960 | 0.866512 |
| Equalized Odds | 0.097950 | 0.026676 | 0.025532 | 0.045989 |

Table 10: A comparison of Fairlearn, and OxonFair when enforcing Equalized Odds. We enforce fairness at 2% on validation to roughly match accuracy with Fairlearn. At this value, there is limited change between versions of Oxonfair and the multi-threshold approach obtains half the fairness violation of Fairlearn at similar accuracy.

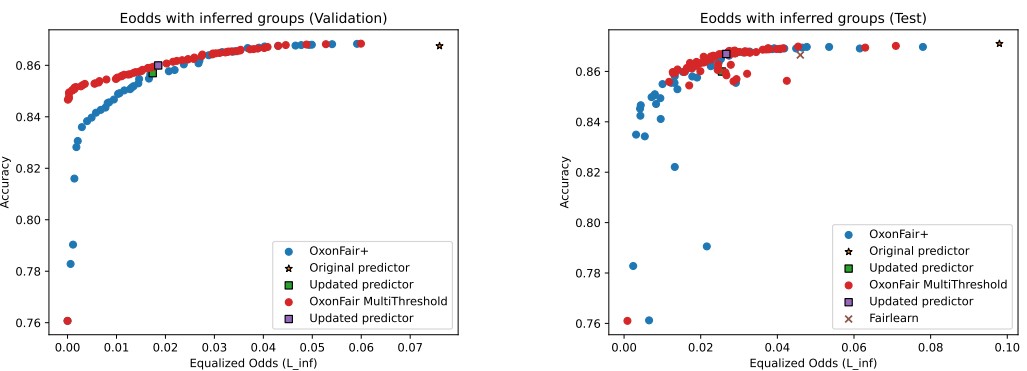

Figure 13: Enforcing Equalized odds using inferred characteristics. While the multi-threshold approach shows clear advantages on the validation set, this does not generalize to unseen test data. For unseen data, single threshold approaches show stronger degradation in accuracy, but their fairness constraints generalize better. This can be attributed to single threshold approaches selecting near constant classifiers when the constraints are strong, while the classifiers found by multi-threshold approaches are more vulnerable to sampling differences between validation and test.

#### D.2.1 Equalized odds without using group membership at test-time

As [57] requires access to groups at test time, we cannot directly compare with [57], we instead compare against FairLearn on the adult dataset, using sex as the protected characteristic. Results can be seen in Table 10, and Figure 13.

Our approach differs from [102], as they enforce fairness with respect to inferred groups (without access to true group labels), while we directly use the inferred group labels to enforce fairness with respect to the underlying true groups.

## E    Computer Vision Experiments

Table 11: Hyperparameter details for the CelebA experiment.

| Hyperparameter | Value/Range |
|---|---|
| Learning Rate | 0.0001 |
| Batch Size | 32 |
| Dropout Rate | 0.5 |
| Backbone | Resnet-50 |
| Weight Decay | 0 |
| Optimizer | Adam [83] |
| Epochs | 20 |

Table 12: CelebA Attribute-level information from Ranaswamy et al. [28]. The columns are target attribute name, percentage of positive samples, skew. For example, Earrings has a skew of 0.97 towards $g=-1$, that is, 97% of positive Earrings samples have gender expression label $g=-1$ (Female)

| Attribute type | Attribute statistics | | |
|---|---|---|---|
| Inconsistently labeled | Positive | Skew | |
| BigLips | 24.1% | 0.73 | $g=-1$ |
| BigNose | 23.6% | 0.75 | $g=1$ |
| OvalFace | 28.3% | 0.68 | $g=-1$ |
| PaleSkin | 4.3% | 0.76 | $g=-1$ |
| StraightHair | 20.9% | 0.52 | $g=-1$ |
| WavyHair | 31.9% | 0.81 | $g=-1$ |
| Gender-dependent | Positive | Skew | |
| ArchedBrows | 26.6% | 0.92 | $g=-1$ |
| Attractive | 51.4% | 0.77 | $g=-1$ |
| BushyBrows | 14.4% | 0.71 | $g=1$ |
| PointyNose | 27.6% | 0.75 | $g=-1$ |
| RecedingHair | 8.0% | 0.62 | $g=1$ |
| Young | 77.9% | 0.66 | $g=-1$ |
| Gender-independent | Positive | Skew | |
| Bangs | 15.2% | 0.77 | $g=-1$ |
| BlackHair | 23.9% | 0.52 | $g=1$ |
| BlondHair | 14.9% | 0.94 | $g=-1$ |
| BrownHair | 20.3% | 0.69 | $g=-1$ |
| Chubby | 5.8% | 0.88 | $g=1$ |
| EyeBags | 20.4% | 0.71 | $g=1$ |
| Glasses | 6.5% | 0.80 | $g=1$ |
| GrayHair | 4.2% | 0.86 | $g=1$ |
| HighCheeks | 45.2% | 0.72 | $g=-1$ |
| MouthOpen | 48.2% | 0.63 | $g=-1$ |
| NarrowEyes | 11.6% | 0.56 | $g=-1$ |
| Smiling | 48.0% | 0.65 | $g=-1$ |
| Earrings | 18.7% | 0.97 | $g=-1$ |
| WearingHat | 4.9% | 0.70 | $g=1$ |
| **Average** | 24.1% | 0.73 | |

### E.1 Methods

We extensively used the codebase of Wang et. al [41] to conduct comparative experiments[11].

- **Empirical Risk Minimization (ERM) [103]:** Acts as a baseline in our experiments where the goal is to minimize the average error across the dataset without explicitly considering the sensitive attributes.

- **Adversarial Training with Uniform Confusion [74]:** The goal is to learn an embedding that maximizes accuracy whilst minimizing any classifier's ability to recognize the protected class. The uniform confusion loss from Alvi et al. [74] is used following the implementation of [41].

- **Domain-Discriminative Training [41]:** Domain information is explicitly encoded and then the correlation between domains and class labels is removed during inference.

- **Domain-Independent Training [41]:** Trains a different classifier for each attribute where the classifiers do not see examples from other domains.

- **OxonFair + Multi-Head [21]:** Described in Section 4.2. $N - 1$ heads are trained to minimize the logistic loss over the target variables, where $N$ is the total number of attributes. A separate head minimizes the squared loss over the protected attribute *Male*. Fairness is enforced on validation data with two separate optimization criteria. **OxonFair-DEO** calls `fpredictor.fit(gm.accuracy, gm.equal_opportunity, 0.01)` to enforce Equal Opportunity. **OxonFair-MGA** calls `fpredictor.fit(gm.min_accuracy.min, gm.accuracy, 0)`.

#### E.1.1 Compute Details

Computer vision experiments were conducted using a NVIDIA RTX 3500 Ada GPU with 12GB of RAM.

Table 13: Comparing accuracy of fairness methods while varying minimum recall level thresholds, $\delta$.

| CelebA - 26 Attributes | $\delta = 0.50$ | $\delta = 0.75$ | $\delta = 0.85$ | $\delta = 0.90$ | $\delta = 0.95$ |
|---|---|---|---|---|---|
| Baseline (ERM) | 89.0 | 84.5 | 80.6 | 77.6 | 72.7 |
| Adversarial | 87.8 | 82.4 | 78.2 | 75.2 | 69.3 |
| Domain-Dependent | 82.3 | 76.8 | 72.4 | 68.6 | 62.2 |
| Domain-Independent | 89.2 | 86.2 | 82.9 | 79.8 | 74.4 |
| OxonFair | **89.9** | **87.3** | **84.4** | **81.8** | **76.9** |

---

[11]`https://github.com/princetonvisualai/DomainBiasMitigation`

Table 14: Extended Version of Table 2. Performance Comparison of Different Algorithmic Fairness Methods on the CelebA Test Set. Results monitor the mean Accuracy, Difference in Equal Opportunity (DEO) and the Minimum Group Minimum Label Accuracy across the attributes.

|  | ERM | Uniconf. Adv [74] | Domain Disc. [41] | Domain Ind. [41] | OxonFair DEO | OxonFair MGA |
|---|---|---|---|---|---|---|
| **Gender-Independent Attributes** | | | | | | |
| Acc. | **93.1** | 92.7 | 93.0 | 92.6 | 92.8 | 90.9 |
| Min grp. min acc. | 64.1 | 72.3 | 76.5 | 71.2 | 72.3 | **85.8** |
| DEO | 16.5 | 19.6 | 14.6 | 7.78 | **3.21** | 3.52 |
| **Gender-Dependent Attributes** | | | | | | |
| Acc. | **86.7** | 86.1 | 86.6 | 85.6 | 85.8 | 82.3 |
| Min grp. min acc. | 43.4 | 53.7 | 59.6 | 53.8 | 52.5 | **78.5** |
| DEO | 26.4 | 25.0 | 21.9 | 6.50 | **3.92** | 3.96 |
| **Inconsistently Labelled Attributes** | | | | | | |
| Acc. | 83.0 | 82.5 | **83.1** | 82.3 | 82.1 | 79.2 |
| Min grp. min acc. | 36.1 | 43.0 | 50.2 | 42.7 | 44.3 | **69.5** |
| DEO | 21.9 | 29.1 | 25.3 | 17.2 | **2.36** | 4.86 |

Table 15: Performance comparison of Baseline, Adaptive g-SMOTE, g-SMOTE, OxonFair-DEO, and OxonFair-MGA on the training set. Reported are the means over the 32 labels selected by [17]. Methods marked * are reported from Zietlow et al. [17].

| 4 Protected Groups | | ERM | Adaptive g-SMOTE [17] | g-SMOTE* [17] | OxonFair-DEO | OxonFair-MGA |
|---|---|---|---|---|---|---|
| **Full Training Set** | Acc. | **90.49** | 85.77 | 87.27 | 89.21 | 86.18 |
| | Min. grp. acc. | 61.74 | 68.06 | 61.84 | 54.20 | **78.48** |
| | DEO | 24.70 | 12.27 | 21.91 | **3.93** | 5.58 |

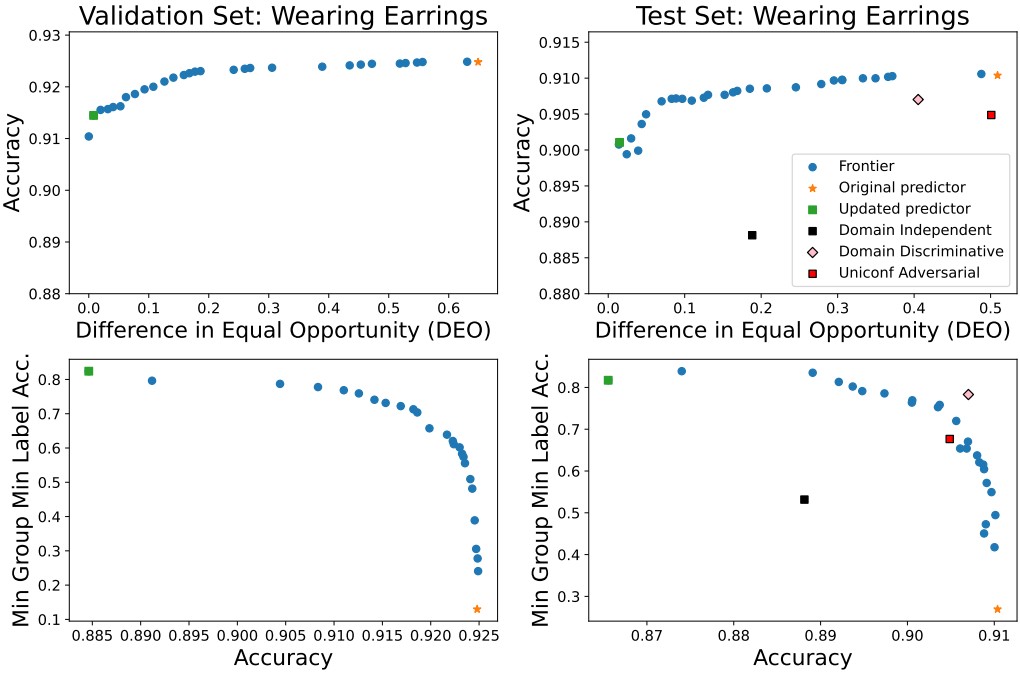

Figure 14: A comparison of the Pareto frontier on validation and test data when enforcing two fairness measures (DEO and Min Group Min Label Acc) for the Wearing Earrings attribute in CelebA whilst monitoring model accuracy.

|  | Gender | Country | Ethnicity | Age |
|---|---|---|---|---|
| English | 41200/7008/6927 | 44487/7744/7639 | 40731/6954/6845 | 39003/6628/6608 |
| Polish | 11782/1461/1446 | 2218/489/471 | 8567/1199/1235 | 8610/1199/1235 |
| Spanish | 2240/407/410 | 2299/436/439 | 2244/407/410 | 2249/407/410 |
| Portuguese | 1408/150/163 | 1105/198/197 | 1377/150/163 | 1389/150/163 |
| Italian | 2730/350/369 | 3769/514/516 | 2706/348/368 | 2676/349/368 |

Table 16: Multilingual Twitter corpus train/val/test statistics.

| | Gender=0 | Gender=1 | Age=0 | Age=1 | Country=0 | Country=1 | Ethnicity=0 | Ethnicity=1 |
|---|---|---|---|---|---|---|---|---|
| English | 5230/14461; 1096/3010; 1074/3009 | 6856/18776; 1447/3917; 1459/3999 | 4937/13279; 1060/2796; 1056/2753 | 6550/18199; 1357/3811; 1341/3874 | 4033/10764; 851/2218; 846/2226 | 7855/13931; 1693/2905; 1687/2996 | 5901/14297; 1236/2962; 1198/2990 | 6036/18614; 1272/3883; 1316/3963 |
| Polish | 370/3552; 172/716; 194/787 | 18/3254; 13/730; 29/674 | 215/2401; 113/505; 103/526 | 18/3248; 14/730; 31/673 | 0/1127; 0/234; 0/253 | 5/629; 2/151; 7/153 | 219/2401; 117/505; 113/526 | 14/3248; 10/730; 21/673 |
| Spanish | 394/997; 102/251; 84/210 | 362/903; 71/159; 77/197 | 354/997; 93/251; 83/210 | 402/903; 80/159; 78/197 | 505/639; 110/155; 112/130 | 213/556; 68/100; 49/118 | 409/997; 100/251; 90/210 | 347/903; 73/159; 71/197 |
| Portuguese | 128/682; 24/60; 13/76 | 18/134; 55/103; 26/74 | 123/682; 29/60; 15/76 | 23/134; 50/103; 24/74 | 34/289; 24/40; 15/52 | 56/39; 50/68; 20/38 | 116/682; 48/60; 19/76 | 30/134; 31/103; 20/74 |
| Italian | 263/1127; 63/273; 63/244 | 119/541; 19/96; 23/106 | 209/1123; 49/272; 42/243 | 171/541; 33/96; 44/106 | 100/748; 19/178; 19/169 | 389/367; 70/65; 83/71 | 377/1121; 81/272; 86/242 | 3/541; 1/96; 0/106 |

Table 17: Multilingual Twitter corpus breakdown. We report the count of positive and total samples across the train/val/test partitions for each ethnicity with specific values. We exclude samples where the label is marked as 'None' for a particular ethnicity.

# F  NLP Experiments

## F.1  Experimental Details

We employ a BERT-based model architecture [86], augmented with an additional head to simultaneously predict demographic factors (see Section 4.2. During training, we utilize the standard cross-entropy loss for the primary prediction task and a mean squared error loss for the demographic predictions, aggregating these to compute the overall loss. We ensure data consistency by excluding entries with missing demographic information. To facilitate easy comparison with different models, we select the Polish language for the multilingual Twitter corpus, noted for its high DEO score, to demonstrate how various models can reduce this score. We also conducted our experiment on the Jigsaw data. Unlike the multilingual Twitter corpus, the Jigsaw religion dataset contains three groups: Christian, Muslim, and others. The entire model, including the BERT backbone, is fine-tuned for 10 epochs using an initial learning rate of $2 \times 10^{-5}$, following the original BERT training setup. All experiments are conducted on an NVIDIA A100 80GB GPU.

## F.2  Hate Speech Detection Task

We follow the methodology outlined in [84] to conduct the hate speech detection task using our tool. Variables such as age and country in the multilingual Twitter corpus are binarized using the same method as described in [84]. The data splits for training, development, and testing are shown in Table 16.

**Multilingual Experiment.** To demonstrate the capability of our proposed tool in handling multilingual scenarios, we conduct experiments across five languages: English, Polish, Spanish, Portuguese, and Italian and the results are shown in Table 18. Observations from the results indicate that: 1) Our model improves equal opportunity performance with minimal sacrifice to the main task performance. 2) The datasets in Polish and Portuguese show higher equal opportunity, indicating more severe bias compared to other languages, yet our proposed method effectively enhances performance in these conditions.

|            | original DEO | updated DEO | original Accuracy | updated Accuracy |
|------------|-----------|----------|-------------------|------------------|
| English    | 5.13      | 3.19     | 84.0              | 84.2             |
| Polish     | 21.4      | 10.1     | 89.6              | 85.8             |
| Spanish    | 9.39      | 1.64     | 69.8              | 67.3             |
| Portuguese | 17.3      | 1.29     | 60.7              | 52.1             |
| Italian    | 7.77      | 0.42     | 75.6              | 77.5             |

Table 18: Multilingual Experiment.

|           | original DEO | updated DEO | original Accuracy | updated Accuracy |
|-----------|-----------|----------|-------------------|------------------|
| Gender    | 21.4      | 8.45     | 89.6              | 88.5             |
| Country   | 10.2      | 8.32     | 81.4              | 82.2             |
| Ethnicity | 8.56      | 4.92     | 83.1              | 82.7             |
| Age       | 12.5      | 6.02     | 82.1              | 80.5             |

Table 19: Demographic Experiments.

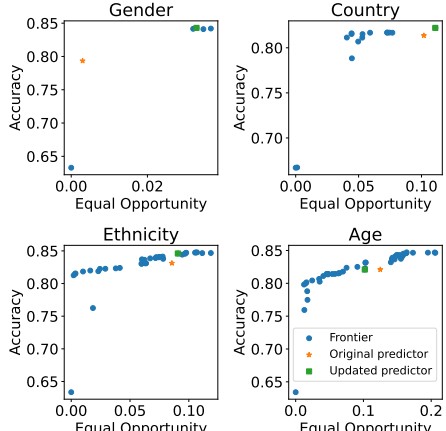

Figure 15: Demographics frontier plot.

**Demographic Experiments.** To demonstrate our tool's ability to address various demographic factors in text, we conducted experiments focusing on age, country, gender, and ethnicity, with results detailed in Table 19 and Figure 15. The outcomes reveal that our tool effectively improves equal opportunities across all demographic factors, underscoring its capability to handle general debiasing scenarios.

|       | Christian    | Other    | Muslim    |
|-------|--------------|----------|-----------|
| Train | 22845/1892   | 3783/554 | 9527/2390 |
| Valid | 5681/470     | 946/148  | 2425/578  |
| Test  | 2944/251     | 604/78   | 1119/319  |

Table 20: Jigsaw religion data.

|       | Black     | Asian    |
|-------|-----------|----------|
| Train | 6718/2811 | 2187/246 |
| Valid | 1684/698  | 547/61   |
| Test  | 841/364   | 284/25   |

Table 21: Jigsaw race data.

### F.3 Toxicity Classification Task

We also evaluate toxicity classification using the Jigsaw toxic comment dataset [85], which has been transformed into a Kaggle challenge. To demonstrate the ability of OxonFair to handle multiple protected groups, we consider religion as the protected attribute and evaluate performance across three groups: Christian, Muslim, and Other. Owing to the limited dataset size, all samples labelled as a religion that was neither Christian nor Muslim were merged into Other and unlabeled samples were discarded. The statistics for this dataset are shown in Table 20, where each cell displays the count of negative and positive samples, respectively. The experimental results are discussed in the main paper.

For the Jigsaw dataset, we follow the setup of [78], selecting race as the protected attribute. We focus on the subset of comments identified as Black or Asian, as these two groups exhibit the largest gap in the probability of being associated with toxic comments. The data statistics are shown in Table 21 where each cell displays the count of negative and positive samples, respectively. The experimental results, presented in Table 23, demonstrate that our proposed tool outperforms all other models.

|       | Religion=Muslim | Religion=Christian | Religion=Other |       | Race=Black | Race=Asian |
|-------|-----------------|--------------------|----------------|-------|------------|------------|
| train | 2390/11917      | 1892/24737         | 554/4337       | train | 2811/9529  | 246/2433   |
| test  | 319/1438        | 251/3195           | 78/682         | test  | 364/1205   | 25/309     |
| valid | 578/3003        | 470/6151           | 148/1094       | valid | 698/2382   | 61/608     |

Table 22: Jigsaw corpus breakdown. We report the count of positive and total samples across the train/val/test partitions for each religion and race with specific values. We exclude samples where the label is marked as 'None' for a particular ethnicity.

|                              | F1 score | Balanced Accuracy | Accuracy | DEO  |
|------------------------------|----------|-------------------|----------|------|
| Base                         | 53.4     | 68.9              | 72.1     | 23.7 |
| CDA [29]                     | 52.7     | 68.2              | 76.4     | 7.65 |
| DP [23]                      | 47.4     | 64.6              | 72.6     | 4.35 |
| EO [3]                       | 47.1     | 64.5              | 73.2     | 5.85 |
| Dropout [88]                 | 52.4     | 68.0              | 72.0     | 12.7 |
| Rebalance [11]               | 51.7     | 67.5              | 74.4     | 5.57 |
| OxonFair (Accuracy)          | 37.5     | 60.8              | 77.7     | 2.10 |
| OxonFair (F1)                | 52.8     | 68.5              | 69.2     | 11.9 |
| OxonFair (Balanced Accuracy) | 52.7     | 68.5              | 68.5     | 0.41 |
| OxonFair (Accuracy)          | 38.6     | 61.1              | 77.5     | 12.3 |
| OxonFair (F1)                | 53.0     | 68.7              | 69.4     | 16.4 |
| OxonFair (Balanced Accuracy) | 53.2     | 68.9              | 67.8     | 20.5 |

Table 23: Jigsaw dataset: Race (w groups: Black, Asian).

# G Comparison Table Information

In this section, we provide further details on the information from Figure 1. While all approaches have many fairness definitions that can be computed, very few can be enforced via bias mitigation. As a minimum, OxonFair supports enforcing the methods from tables 5 and 6 (eliminating duplicates give the number 14 in the table). In addition to this, it supports a wide range of metrics that aren't used in the literature, for example minimizing the difference in balanced accuracy, F1 or Matthews correlation coefficient (MCC) between groups, e.g., by using `balanced_accuracy.diff` as a constraint. It also supports the definitions set out in Appendix C, including minimax notions; absolute bias amplification; and enforcing for minimum rates per group in recall, or precision, or sensitivity actively promoting *levelling-up* [8].

## G.1 FairLearn Methods Support

Fairlearn provides an overview of the supported bias mitigation algorithms and supported fairness constraints in their documentation[12]. The number of performance and fairness objectives supported are dependent on the method.

Methods supported include ExponentiatedGradient and GridSearch that provide a wrapper around the reductions approach to fair classification of Agarwal et al. [31]. Supported fairness definitions for classification are Demographic-Parity, Equalized Odds, True Positive Rate Parity, False Positive Rate Parity and Error Rate Parity. For postprocessing the ThresholdOptimizer approach of Hardt et al. [3] is supported. The adversarial approach of [32] is also supported and can enforce fairness based on Demographic Parity and Equalized Odds. The CorrelationRemover method provides preprocessing functionality to remove correlation between sensitive features and non-sensitive features through linear transformations. It should be emphasized that Fairlearn also provides an interface for defining custom Moments for fairness and objective optimization, however, as of the current version 0.10 no documentation or examples are provided for doing so.

## G.2 AIF360 Methods Support

AIF360 provides support for a wide variety of methods[13][14] that enforce fairness, many of which overlap with Fairlearn. We consider group fairness approaches.

Preprocessing algorithms include DispirateImpactRemover [11], LFR [104], Optimized Preprocessing [26], Reweighting [25] and FairAdapt [105]. Inprocessing algorithms include AdversarialDebiasing [32], PrejudiceRemover [106], Exponentiated GradientReduction and GridSearchReduction [31]. Postprocessing approaches include CalibratedEqOddsPostprocessing [44], EqOddsPostprocessing [3], RejectOptionClassification [25].

## G.3 Societal Impacts

We reiterate the findings of Balayn et al., who note that fairness toolkits can act as a double-edged sword [92]. Open source toolkits can enable wider adoption of the assessment and mitigation of bias and fairness related harms. However, if misused, these toolkits can create a flawed certification of algorithmic fairness, endangering false confidence in flawed methodologies [55, 107]. We join growing calls in encouraging practitioners to be reflective in their use of fairness toolkits [60]. Specifically, we urge practitioners to adopt a harms first approach to fairness and be reflective in their measurement and enforcement of fairness.

---

[12]`https://FairLearn.org/main/user_guide/mitigation/index.html`
[13]`https://aif360.readthedocs.io/en/stable/modules/algorithms.html`
[14]`https://aif360.readthedocs.io/en/stable/modules/sklearn.html`

