# OpenReview forum: "OxonFair: A Flexible Toolkit for Algorithmic Fairness"
_NeurIPS.cc/2024/Conference — NeurIPS 2024 poster_

### Official Review · Reviewer_aniZ · 2024-07-10

**Soundness:** 3
**Presentation:** 2
**Contribution:** 3
**Rating:** 7
**Confidence:** 3

**Summary:**

The paper introduces "AnonFair," a toolkit designed to enforce algorithmic fairness across various domains, including NLP, computer vision, and traditional tabular data. It is compatible with popular machine learning frameworks like sklearn, AutoGluon, and PyTorch. Unlike well-established fairness tools like FairLearn and AIF360, AnonFair extends to different types of data, including NLP and vision.

Other tools offer many methods but limited control over them, while AnonFair uses a single, highly customizable method that allows for per-group thresholding.

It specifically addresses the issue of overfitting by utilizing validation data, making it more reliable when traditional methods might fail.

Empirical evidence presented shows that AnonFair performs well, often matching or surpassing other methods in fairness benchmarks without being specifically optimized for complex or high-dimensional scenarios.

AnonFair seems to provide a robust and adaptable solution for implementing fairness in machine learning, in ways that other tools do not currently offer.

**Strengths:**

- The paper does well in positioning AnonFair against competing tools by demonstrating its performance on standard fairness metrics and its versatility across a variety of use cases.
- AnonFair supports NLP and computer vision classification tasks, allowing broader applicability.
- The toolkit uses validation data to combat overfitting, ensuring that fairness measures remain robust across both training and unseen data.

- The toolkit not only competes well in terms of accuracy and fairness metrics but also offers significant advantages in computational efficiency.

**Weaknesses:**

- Some sections are overly detailed, such as the introduction, while others are missing necessary depth:
    - Section 3 could use a clearer structure, possibly with a diagram, to help readers understand how to interact with the toolkit.
    - The section on toolkit expressiveness needs more detailed examples and explanations of how the supported fairness measures are implemented.
    - Results discussion is kept very brief and could benefit from specific numerical examples, like percentage improvements compared to other methods.m actual numbers, such as how much % improvement in comparison to method XY and such.

- The paper assumes readers are familiar with fairness terminology and metrics without adequate explanations or definitions for some acronyms (e.g., DEO in Table 3 and 4).
    - Subsection 4.3 lists supported fairness measures but fails to provide examples or brief explanations, making it less informative for those not familiar with these terms.

- Lack of consistency in terminology usage; for example, "EOp" in Figure 1 (top right) vs. "EO" in Section 5.2, “AnonFair” missing before "Frontier" in Figure 1 (left), and inconsistent references like "See Figure" vs. "See fig.."

- A stronger call to action for community engagement, such as through open-source collaboration or empirical validation studies, could significantly enhance the broader impact and encourage more widespread adoption and refinement of AnonFair.

- The paper would benefit from a summary of explicit cases and recommendations advising users on the best scenarios for using the tool.

- Figure 2 is not referred to in the paper, or did I miss this part.

**Questions:**

1. The paper mentions that hard assignment is more efficient than soft assignment, while appendix A adds some operational details, it remains unclear how these methods specifically compare in terms of quantitative metrics. Could the authors provide specific metrics or comparisons that demonstrate the efficiency and performance benefits of hard assignment?
2. The discussed limitations reads a bit out of context given provided evidence in the paper. What makes the mentioned  solutions suboptimal, and how significant are these shortcomings?  Also it was not clear to me, after finishing reading,  when it is adequate to use this tool and what could be use cases when it fails. Including this into the conclusion could make the reader grasping the full picture.
3. Is Figure 6 part of the Appendix or misplaced?

**Limitations:**

Some of the limitations are acknowledged, but could be expanded with more actionable insights.
A call to action for community engagement, such as through open-source collaboration would also encourage broader impact and adoption of AnonFair against its competitors.
It would be beneficial if the authors suggested potential improvements or future research directions for the suboptimal fairness metrics and data scarcity issues mentioned.
The broader impact section identifies ethical concerns well. However, detailing the intended applications and scenarios where AnonFair might be most effective, or where it could fail, would provide readers and users with clearer guidance on its practical use and limitations.

---

> ### Author Rebuttal · Authors · 2024-08-05
>
> Thank you for taking the time to review our manuscript and for providing detailed, helpful, and constructive feedback. We hope to address outstanding weaknesses and concerns below.
>
> **Improving presentation:** The idea of a figure/flow chart is a good one, but there is insufficient space in the paper. We will add this directly to the toolkit documentation instead, where it is more likely to be seen by someone interested in using the toolkit.
>
> **Details on implementing fairness measures:** Please see Appendix B for details of how measures are implemented. In brief, you can just write down a function of the confusion matrix. This is computed per group, and standard measures such as minimum per group, or average difference or ratio of measures between groups are automatically computed. These correspond to different notions of fairness. We will add an additional example of this for Equal Opportunity, as this is one of the most widely used group fairness measures.
>
> **Results discussion and numerical examples:** We will discuss specific examples from the tables and figures where our toolkit shows clear improvements over other approaches.
>
> **Improving the consistency of fairness terminology and abbreviations:** Thanks for flagging the inconsistency. We will correct this.
>
> **Call to action for community engagement:** Thank you. We will add a call for community engagement in the abstract. We strongly agree with the reviewer’s focus on making the toolkit more accessible through documentation. Unfortunately, space constraints mean that much of the information is going to lie outside the paper. We have spent much of the past months since submission extending the documentation and adding additional examples. This will continue to be a focus going forward.
>
> We invite community-driven contributions principally as pull requests reflecting the needs and considerations that researchers come across in practice. In this manner, we can support more advanced features without overwhelming the codebase and maintenance requirements. Our toolkit is now a pip installable library, and the community can raise issues, concerns, and requests on a GitHub repository. In this way, practitioners can collaboratively build on each other's work.
>
> **Explicit cases and recommendations advising users on best practices. What are the adequate uses of this tool and what could be the use cases when it fails?**
>
> These are very important questions. The short answer is that this is too important to be compressed into the conclusion of the paper.
>
> This toolkit arose from a promise to a healthcare provider that we would go through existing toolkits and return a list of best practices. Essentially, our conclusion was, it didn’t matter what we suggested as no existing toolkit worked for NLP or computer vision, and those that existed for tabular data couldn’t enforce measures like bias amplification, conditional measures, or levelling up. The purpose of this paper is to lay out a technical toolkit which is expressive enough to support both our recommended best practices, and hopefully the best practices of other researchers who disagree with us. Follow-up white papers will set out how we think it should be used. Use cases, best practices and documentation are our core focus going forward.
>
>
> **Figure 2 not referred to in the main paper:** We will add references to this figure in Section 4 (specifically lines 156 and 189 and footnote 5).
>
> # Questions
>
> **Question 1:** For a performance comparison, we reran the unconditional fairness metrics from table 7 with inferred attributes and the slow vs fast pathway. Using default parameters, the fast pathway has an average of 1.2% better fairness across the metrics and wins in 7/10 of the cases, while maintaining similar accuracy (win rate 5/10, an average decrease in 0.28% of accuracy).
>
> For an efficiency comparison, we reran the comparison with fairlearn, on adult, varying the number of groups.
>
> | # Groups  |5|4|3|2|
> |-|-|-|-|-|
> | FairLearn |33.2s|33.3s|25.4s|19.4s|
> |Fast| 0.7+55.7s|0.7+0.93s|0.7+0.078s| 0.7+0.05s|
> |Slow| - | 0.7+454s| 0.7+22s|0.7+1.5s|
>
>
> We will add these numbers to the paper.
>
> **Question 2:** Apologies. The fact that thresholding is suboptimal for Equalized Odds is widely known in the literature [1]. The suboptimality comes from the fact that Equalized Odds is a combination of two fairness measures (difference in true positive rates and difference in false positive rates) and that ideal thresholds for one measure need not minimize the other. Instead, a combination of thresholding and randomization is often used. Our approach is only known to be sub-optimal when the true attributes are used, and not with data that requires the use of inferred attributes. The rest of the time, Anonfair has no guarantees regarding optimality.  This is the same as most widely used fairness methods including FairLearn. Unfortunately, the only answer to if our approach should be used is the same as for the rest of machine learning – the experiments might look good, but where alternatives exist, it is best practice to compare them on your use case.
>
> The only case where our method can be said to fail is where insufficient data is provided to generalize from validation to unseen test data. Otherwise, the method may be suboptimal (better fairness accuracy trade-offs might exist), but it will work, and similar fairness/accuracy trade-offs will be seen on unseen test data.
>
> **Question 3:** Figure 6 is part of the appendix and is an expanded Figure 1 from the main body. While Figure 1 compares fairness toolkits with a focus on optimizing for balanced accuracy, Figure 6 broadens this comparison by including additional metrics.
>
> [1] Hardt, Moritz, Eric Price, and Nati Srebro. "Equality of opportunity in supervised learning." Advances in Neural Information Processing Systems 29 (2016).

---

> ### Author Response · Authors · 2024-08-12
> **Any follow-up clarifications?**
>
> We hope that we have addressed all issues raised to your satisfaction in our rebuttal. We would be happy to provide additional clarifications if required as the discussion period will be over soon.
>
> Thank you for your time.

---

> > ### Comment · Reviewer_aniZ · 2024-08-12
> >
> > I appreciate the authors' detailed response to the concerns raised and the additional experiments.
> > Given the constraints of a conference paper, I agree that the toolkit's documentation may be better suited for the extensive details.
> > With the reviewers' feedback incorporated, the paper should be better positioned to convince practitioners to try out the toolkit. Good luck.
> > I will maintain the current score.

---

### Official Review · Reviewer_JNWw · 2024-07-13

**Soundness:** 3
**Presentation:** 2
**Contribution:** 3
**Rating:** 7
**Confidence:** 1

**Summary:**

This paper describes a new toolkit for algorithmic fairness, enabling the optimization of any fairness measure that is a function of the confusion matrix. Experiments on vision and NLP demonstrated the effectiveness of the proposed toolkit.

**Strengths:**

An easy-to-use toolkit for enforcing algorithmic fairness.

**Weaknesses:**

Presentation could be made more self-contained, e.g. a table listing the supported fairness metrics, as functions of the confusion matrix. This would help readers not familiar with the field.

It seems that only binary classification is supported. How can such metrics be extended to other tasks?

Some minimal code snippets for the interface could be shown as examples.

**Questions:**

- L5: "True positives, false positives, ..." => "the confusion matrix"
 - L6: "extendable" => "extensible"

**Limitations:**

The authors adequately discussed the limitations of their toolkit.

---

> ### Author Rebuttal · Authors · 2024-08-05
>
> We thank the reviewer for the feedback and helpful suggestions that will be integrated to improve the paper.
>
> **Improvements in presentation:** We will add the definition of equal opportunity (the most common fairness definition, corresponding to difference in recall between groups) to the work. Honestly, we have substantial problems with space constraints in this work. The two review papers we cite [1,2] only list and discuss metrics and come in at a combined 15+ double-columned pages. We will try to include the most common or most relevant metrics in our work, but we are not a review paper, and providing references for most of the metrics is probably the most sensible option.
>
> **Beyond binary classification:** Tasks beyond binary classification are on our roadmap for future work. Fair ranking is of particular interest, as failures in ranking have clear well-defined harms and may be compatible with our accelerated approach. For multiclass classification, from a policy perspective, there are a couple of metrics that are generalizations of demographic parity (every labelling rate should be the same for all groups), and equalized odds (confusion matrix should be the same for all groups), but they seem to have been chosen for mathematical convenience, rather than because they correspond to clear harms.
>
> **Minimal Code Snippets:** We agree that minimal code snippets would improve the paper. We have code snippets in the appendix (Appendix C.2 and figure 8). and we will add further examples. In addition to code snippets, we also have Jupyter Notebook tutorials and example studies for practitioners to get up and running with our toolkit. We will highlight the vast availability of resources and examples that our toolkit provides, and our toolkit empowers the community to build on these.
>
>
> **Typos:** Thanks for pointing out typos, these will be promptly corrected.
>
> [1] Verma, Sahil, and Julia Rubin. "Fairness definitions explained." Proceedings of the international workshop on software fairness. 2018.
>
> [2] Hardt, Michaela, et al. "Amazon sagemaker clarify: Machine learning bias detection and explainability in the cloud." Proceedings of the 27th ACM SIGKDD conference on knowledge discovery & data mining. 2021.

---

> ### Author Response · Authors · 2024-08-12
> **Any follow-up clarifications?**
>
> We hope that we have addressed all issues raised to your satisfaction in our rebuttal. We would be happy to provide additional clarifications if required as the discussion period will be over soon.
>
> Thank you for your time.

---

> ### Comment · Reviewer_JNWw · 2024-08-13
>
> Thanks for the detailed response. I have modified the score based on the appendix. Thanks.

---

### Official Review · Reviewer_AiSk · 2024-07-13

**Soundness:** 3
**Presentation:** 3
**Contribution:** 3
**Rating:** 6
**Confidence:** 4

**Summary:**

The paper introduces a new toolkit designed to enhance algorithmic fairness with greater expressiveness. Unlike existing toolkits, this one offers more customization options to optimize user-defined objectives and fairness constraints. Although the proposed toolkit currently includes only one method, it supports both computer vision and natural language processing (NLP) tasks. The authors compare the efficiency of this method, finding that the toolkit is relatively more efficient than Fairlearn. Comprehensive experiments were conducted on various datasets, and the results were compared with those from other popular toolkits.

**Strengths:**

- The paper introduces a versatile toolkit that supports both NLP and computer vision tasks, unlike existing toolkits which lack this capability.
- The proposed toolkit employs efficient optimization techniques that accelerate the evaluation process.

**Weaknesses:**

- The formulation presented in Subsection 4.2 of the paper is limited to a single-layer model, which restricts its applicability across different machine learning models. To enhance the flexibility of the method, I recommend adopting a more generic notation, particularly if we aim to incorporate pretrained language models.
- The abstract is quite unclear, especially the part that mentions "9/9 and 10/10 of the group metrics of two popular review papers." I suggest rephrasing the abstract for better clarity and comprehension.

**Questions:**

- In Figure 3, the proposed toolkit appears to encounter scaling issues when reaching 5 groups. Could you provide more details on why this occurs and elaborate on the underlying reasons for this limitation?
- The paper presents results on multilingual datasets. Do you have any specific findings for each language, particularly regarding the effectiveness of the toolkit for individual languages?

**Limitations:**

Yes

---

> ### Author Rebuttal · Authors · 2024-08-05
>
> We thank the reviewer for their detailed feedback. We are happy to see that the reviewer appreciates the versatility of our toolkit beyond existing solutions and the efficiency of optimization in the toolkit.
>
> **Clarification on notation:** The equations in section 4.2 contain a function, $B(x)$. This represents an arbitrary non-linear backbone shared between the two heads. While the heads are linear (as is common in multitask learning), arbitrary non-linear functions can be learned. This approach is used as written for the NLP and Computer Vision tasks which use complex non-linear backbones. We will emphasize this in the text.
>
> **Improvements to Abstract:** Thanks. We will rephrase the abstract to improve clarity.
>
> # Questions
>
> > In Figure 3, the proposed toolkit appears to encounter scaling issues when reaching 5 groups. Could you provide more details on why this occurs and elaborate on the underlying reasons for this limitation?
>
>
> The slowdown as we increase the number of groups, is expected and supported by the analysis in Section 4 (lines 157-162) which shows that the run-time is exponential in the number of groups. This was a conscious decision to maximize expressibility at the expense of compute for a large number of groups. Having sufficient data to enforce fairness for a large number of protected groups is unfortunately rare in algorithmic fairness and in practice, these scaling issues are rarely encountered.
>
> >The paper presents results on multilingual datasets. Do you have any specific findings for each language, particularly regarding the effectiveness of the toolkit for individual languages?
>
> Our toolkit is language and data modality agnostic. While we see differences in performance, this seems to be predominantly driven by the quality of the data. In general, the most common languages have much more data available (particularly  English with three times more data than the next most available language), and this allowed curators to create datasets with a balanced subset of groups, and positive and negative labels. For example, in the case of well-resourced languages such as English (.499 of datapoints were labelled Female) and (.495 of English datapoints were labelled non-White). In contrast for Polish protected groups were very unbalanced (.324 Female) and (.105) non-White.
>
> This means both that base classifiers for well-resourced languages tend to be fairer (at least when evaluated on the provided test set) but also that the estimation of statistics such as recall are more stable, and our fairness toolkit generalizes better to the test set. In Appendix E.2 Multilingual Experiment (from line 813), we find that the base classifiers in Polish and Portuguese show higher difference in equal opportunity, indicating more severe bias compared to other languages (e.g., English). Where there is such a large initial bias, our toolkit can make larger improvements in fairness, even  if the underlying statistics cannot be reliably estimated.

---

> > ### Comment · Reviewer_AiSk · 2024-08-08
> >
> > Thank you for the response and plan to improve the paper.
> >
> > I will keep my scores.

---

### Official Review · Reviewer_AKgy · 2024-07-14

**Soundness:** 3
**Presentation:** 2
**Contribution:** 2
**Rating:** 4
**Confidence:** 4

**Summary:**

The paper describes details of a fairness toolkit ("AnonFair"), which confers fairness to any given machine learning classifier by exploring a wide range of prediction thresholds for different groups (which are either provided upfront or inferred through an auxiliary classifier). The toolkit is designed to be quite expressive, as it can optimize several different metrics, e.g., false positives/negatives, true positives, etc. The toolkit can work across all classifiers (which can output class probabilities), including ones trained on vision and NLP tasks.

**Strengths:**

The paper introduces and describes a toolkit that implements several fairness strategies and can support any fairness measure that can be expressed in terms of true positives, false positives, true negatives and false negatives. These techniques primarily rest upon adjusting the classification thresholds of different groups, and the paper also incorporates tricks to speed up their computations of precision and recall across different thresholds. The fairness techniques that this paper implements are (largely) classifier agnostic, and can be applied to a wide range of classifiers including NLP and vision classifiers (as this paper shows). Overall, I appreciate that expressivity and broad applicability of their toolkit.

**Weaknesses:**

While the toolkit might turn out to be useful for some practitioners, it is a relatively straightforward implementation of well-known (and simple) technique of adjusting prediction thresholds across groups. Exploring different thresholds can be computationally prohibitive, for which the authors use a standard trick to speed up their explorations (which I appreciate). The paper acknowledges and cites relevant papers/techniques that they implement.   Overall, the originality and novelty of their work is significantly limited, as the toolkit is an implementation of known and simple fairness techniques. Further, the underlying fairness techniques (not from the authors) are themselves applicable to most classifiers, so any implementation of the same could work for NLP and vision tasks—which is claimed to be one of the major contributions of this work.

**Questions:**

I feel that the current version is a good starting point (in terms of implementation) of existing fairness techniques and speeding them up and trying them out on vision and NLP tasks. To improve the paper, I would suggest clearly outlining the important problems that this toolkit now can enable researchers to answer (which was not possible before) and answer a few of those questions in the paper.

**Limitations:**

I believe the paper adequately communicates their shortcomings and cites past references when using them. However, I think it might help to also acknowledge that the underlying fairness techniques broadly apply to a wide range of classifiers, and naturally extend to classifiers in computer vision and NLP domains. Reading parts of the paper felt like that there are significant challenges in adoption of fairness techniques to NLP and CV, and this paper overcomes them through novel solutions—which is not the case.

---

> ### Author Rebuttal · Authors · 2024-08-05
>
> We thank the reviewer for their review, and we hope to address the issues raised.
>
> ­In brief, there are two main issues we wish to discuss. 1. what this toolkit does 2. The limited novelty of any toolkit/library, and that such libraries are explicitly covered in the call for papers.
>
> 1. We are concerned that the review substantially understates what the toolkit does. While our toolkit is a post-processing method, it does not simply ‘set per group thresholds’ and works when group annotations are unavailable at test time. To this end, we offer two modifications:
>      (a) A classifier agnostic approach that uses an auxiliary classifier to estimate groups, while enforcing fairness for the true groups – and not the estimated groups.
>      (b) Model surgery for neural networks (section 4.2, based on work by  [4]). This involves training a multi-headed network that jointly estimates groups alongside the original task. These heads are then merged, resulting in a single fair network with one head and the same architecture as the original network.
>      Both approaches have only been previously shown to work for demographic parity [4,5] and not for any other fairness measure.
>
> 2. The NeurIPS 2024 Call for Papers explicitly calls for “libraries, improved implementation and scalability” under the infrastructure theme. Software libraries have been published in the NeurIPS main track [1-3]. Like any good toolkit we prefer well-tested and understood components, over speculative new methods. Rather than rehashing an existing argument, we refer the reviewer to this publicly available debate on Openreview from an accepted Neurips toolkit paper last year [1].
>
> ­We still believe that novelty is important, but novelty should be evaluated with respect to other toolkits, and ask what can this work do that other toolkits cannot?
>
> Compared to others, this is the only toolkit that works on computer vision and NLP, and is substantially more expressive than anything else out there. This generalization to NLP, and computer vision, works without group membership at test-time, and is not a natural extension of per group thresholding.
>
> Among many other measures, we are the only toolkit that simultaneously supports conditional fairness metrics, minimax fairness, and levelling up through minimum rate constraints. Unlike other toolkits, we jointly and efficiently optimize a performance objective. This not only minimizes degradation while enforcing fairness but can improve the performance of inadequately tuned unfair baselines. Our toolkit is compatible with more frameworks than existing approaches including scikit-learn, Autogluon, and PyTorch.
>
> ­
>
> > To improve the paper, I would suggest clearly outlining the important problems that this toolkit now can enable researchers to answer (which was not possible before) and answer a few of those questions in the paper.
>
> Please see levelling up [6] examples in figure 4, and the adjacent table, figure 8 and appendix C.3. Minimax fairness [7] in appendix C.1, and table 8. Conditional metrics in appendix C.4.  Directional Bias Amplification [8] in appendix C.5. And a whole host of additional measures taken from the fairness review paper [9] in table 7.
>
> # References
>
> [1] Griffiths, Ryan-Rhys, et al. "GAUCHE: a library for Gaussian processes in chemistry." Advances in Neural Information Processing Systems 36 (2023).
>
> [2] Jamasb, Arian, et al. “Graphein-a python library for geometric deep learning and network analysis on biomolecular structures and interaction networks.” Advances in Neural Information Processing Systems 35 (2022): 27153-27167.
>
> [3] Pineda, Luis, et al. “Theseus: A library for differentiable nonlinear optimization.” Advances in Neural Information Processing Systems 35 (2022): 3801-3818.
>
> [4] Lohaus, Michael, et al. "Are two heads the same as one? Identifying disparate treatment in fair neural networks." Advances in Neural Information Processing Systems 35 (2022): 16548-16562.
>
> [5] Aditya Krishna Menon and Robert C Williamson. The cost of fairness in binary classification. In Conference on Fairness, accountability and transparency. PMLR, 2018.
>
> [6] Mittelstadt, Brent, Sandra Wachter, and Chris Russell. "The Unfairness of Fair Machine Learning: Levelling down and strict egalitarianism by default." arXiv preprint arXiv:2302.02404 (2023).
>
> [7] Martinez, Natalia, Martin Bertran, and Guillermo Sapiro. "Minimax pareto fairness: A multi objective perspective." International conference on machine learning. PMLR, 2020.
>
> [8] Wang, Angelina, and Olga Russakovsky. "Directional bias amplification." International Conference on Machine Learning. PMLR, 2021.
>
> [9] Hardt, Michaela, et al. "Amazon sagemaker clarify: Machine learning bias detection and explainability in the cloud." Proceedings of the 27th ACM SIGKDD conference on knowledge discovery & data mining. 2021.

---

> ### Comment · Reviewer_AKgy · 2024-08-10
> **Response**
>
> Thanks for your response. I agree with the authors that their toolkit also supports cases when group annotations are unavailable. I realize that my evaluation might not have properly taken this into account. In that light, I have increased my assessment score about the contribution from poor to fair, and overall assessment from 3 to 4.
>
> I'm quite aware of the fact that NeurIPS CFP invites libraries as contributions, but that doesn't take away my concerns about the (lack of) novelty and originality of the underlying techniques implemented in the library.

---

### Official Review · Reviewer_9LRe · 2024-07-15

**Soundness:** 3
**Presentation:** 2
**Contribution:** 3
**Rating:** 6
**Confidence:** 3

**Summary:**

This paper presents AnonFair, a cutting-edge open-source toolkit designed to promote algorithmic fairness. Authors claim the following contributions:
(1) Comprehensive support for NLP and Computer Vision classification, as well as standard tabular problems.
(2) Enhanced robustness against overfitting challenges through the ability to enforce fairness on validation data.
(3) Versatility in optimizing any measure that is a function of True Positives, False Positives, False Negatives, and True Negatives, making it easily adaptable and more expressive than other toolkits.
(4) Seamless integration with popular ML toolkits such as sklearn, Autogluon, and pytorch.
(5) AnonFair supports 9/9 and 10/10 of the group metrics of two prominent review papers and is accessible online at no cost.

**Strengths:**

This toolkit progresses in algorithmic fairness and enhances multidisciplinary collaborations, it is design to integrate the intervention of policy-makers.

The paper includes a complete section of experiments and comparison with existing toolkits.

AnonFair key contributions include support to popular and relevant NLP and Computer vision areas.

**Weaknesses:**

* Lack of clarity in some reported experiments, e.g. results tables are not cited in the text, metrics are not well-contextualized (e.g. larger or lower scores are better?)

* Lack of analysis, examples or human evaluation to better understand contributions and limitations of the method in each of the experiments.

**Questions:**

(1) Could you provide more high-level context for each of the experiments that you are running in order to make the paper more self-contained?
(2) for NLP experiments, why do you think mitigation works for Twitter and not for Jigsaw?

**Limitations:**

Authors report some limitations, but further analysis on the experiments could raise more limitations that may be currently ignored.

---

> ### Author Rebuttal · Authors · 2024-08-05
>
> We thank the reviewer for the positive comments and constructive feedback.
>
> ---
>
>
> # Additional clarity in presentation
>
> We will use arrows in tables to indicate if larger **(↑)**, or lower **(↓)** scores are better. We will also discuss this when mentioning the different fairness metrics to improve accessibility for readers. For example, when using Difference in Equal Opportunity (a smaller score is better/fairer). To improve presentation, we will also add references to the results table and relevant appendices to the main body of the paper.
>
> ---
>
> # Questions and additional analysis of results
>
>
>
>
> >**Question (1)**: Could you provide more high-level context for each of the experiments that you are running in order to make the paper more self-contained?
>
> We will add text to these experiments.
>
> In brief, the computer vision and NLP experiments (tables 1 through 4), are standard fairness benchmarks where existing methods compete to minimize Equal Opportunity (i.e. difference in recall between groups) while maintaining high accuracy. This corresponds to a situation where high accuracy is important, but you don’t want the burden of low recall to disproportionately fall on particular groups. A scenario where this might be important is medical testing where you want to ensure that there is a similar recall rate for all groups.
>
> Results in the appendices show the expressiveness of the toolkit. We simply want to show that these well-cited measures are optimizable using our approach.
>
> ---
>
> >**Question (2)**: for NLP experiments, why do you think mitigation works for Twitter and not for Jigsaw?
>
> Good question. While we do improve fairness on Jigsaw, it is not as reliable as on Twitter. One of the nice things about our approach is that we directly enforce constraints on validation data and given sufficient data these results should generalize to unseen test data.  This does not happen on Jigsaw with the same reliability we see on Twitter. Largely this can be attributed to data limitations. Equal Opportunity is inherently an unstable measure (it looks at the difference in recall between groups) where both the number of individuals in a particular group is small and the ratio of hate speech is low, we can have very limited data for measuring recall let alone differences in recall.
>
> While limited validation data could be worked around using techniques such as cross-fold validation (this is compatible with our approach, and integrating it is ongoing work), Jigsaw comes with a pre-existing test split, and this was frequently unrepresentative of the combined train and val set. This problem is exacerbated as unlike Twitter, Jigsaw contains scenarios with more than two protected groups.
>
> We will add an appendix with data counts and discuss the limitations clearly there.
>
> ---
>
> >Does this additional analysis raise new limitations?
>
> We briefly touch on the issue of data in our limitations section, but we did not have sufficient space to discuss this in detail. We’ll add text based on the above answer to the appendix and link to it from the experimental section.

---

> ### Author Response · Authors · 2024-08-12
> **Any follow-up clarifications?**
>
> We hope that we have addressed all issues raised to your satisfaction in our rebuttal. We would be happy to provide additional clarifications if required as the discussion period will be over soon.
>
> Thank you for your time.

---

> ### Comment · Reviewer_9LRe · 2024-08-13
>
> Thanks a lot for your responses, it clarifies my questions, I will keep my score.

---

### Author Rebuttal · Authors · 2024-08-05

We thank the reviewers for their helpful and largely positive comments (**overall scores 7,6,6,6,3**). The suggestions are informative, and we will adjust presentation in the paper wherever an issue has been raised.
\
\
Our toolkit provides a “robust and adaptable solution for implementing fairness in machine learning, in ways that other tools do not currently offer” (**reviewer aniZ**). We “include support to popular and relevant NLP and Computer vision areas” (**reviewer 9LRe**) and this is “unlike existing toolkits which lack this capability” (**reviewer AiSk**). We provide “an easy-to-use toolkit for enforcing algorithmic fairness” (**reviewer JNWw**) and reviewers “appreciate the expressivity and broad applicability” of the toolkit (**reviewer AKgy**). We contribute “a complete section of experiments and comparison with existing toolkits” and our toolkit represents “progress in algorithmic fairness and enhances multidisciplinary collaborations” (**reviewer 9LRe**).
\
The score 3 (**reviewer AKgy**) arises from a concern regarding lack of novelty. This is entirely understandable. This is a library submission and as such, most of what we are doing is putting together existing pieces in a useful way and performing good engineering to substantially increase expressiveness and performance (all of which the review acknowledges and appreciates). However, the NeurIPS 2024 Call for Papers explicitly includes “libraries, improved implementation and scalability” under the infrastructure theme. These software libraries have been published in the NeurIPS main track [1-3].


Beyond this, the review misses some of our more substantial contributions. We are not simply adjusting thresholds. All NLP and computer vision experiments are the result of network surgery (see section 4.2), where a new fair network is generated with the same underlying architecture as the base network. These experiments do not use protected attributes at test-time.

Moreover, the review requested additional experiments showing fairness definitions that this toolkit, and no other toolkit can solve.  These can be seen in tables 6,7 and 8, figure 4, and appendices C.1, C.3, C.4, and C.5.

Otherwise, all reviewers are in agreement about our contribution. This is the only fairness toolkit to work for computer vision or NLP and represents a step forward not just in terms of the domains where it has been applied but also in the wide range of fairness constraints that it can solve.


# References

[1] Griffiths, Ryan-Rhys, et al. "GAUCHE: a library for Gaussian processes in chemistry." Advances in Neural Information Processing Systems 36 (2023).

[2] Jamasb, Arian, et al. “Graphein-a python library for geometric deep learning and network analysis on biomolecular structures and interaction networks.” Advances in Neural Information Processing Systems 35 (2022): 27153-27167.

[3] Pineda, Luis, et al. “Theseus: A library for differentiable nonlinear optimization.” Advances in Neural Information Processing Systems 35 (2022): 3801-3818.

---

### Decision · Program_Chairs · 2024-09-25

**Decision:**

Accept (poster)

**Comment:**

The paper presents AnonFair, a new open-source toolkit for enforcing algorithmic fairness that supports NLP and computer vision classification tasks in addition to tabular problems. Such broader applicability to different data types is a key advantage over existing fairness toolkits.

AnonFair can enforce fairness on validation data, making it more robust to overfitting challenges compared to other approaches. It supports 9/9 and 10/10 of the group metrics from two prominent fairness review papers.

AnonFair is compatible with popular ML frameworks like scikit-learn, AutoGluon, and PyTorch.

**Strengths** Broader applicability to NLP and computer vision tasks compared to existing fairness toolkits, with more expressive and customizable fairness measures. It features improved robustness to overfitting by using validation data and competitive performance on standard fairness benchmarks

**Weaknesses**

Limited novelty in the underlying fairness techniques (mainly implements known methods). It currently only supports binary classification tasks
Scaling issues with larger numbers of protected groups



Overall,
this is a solid toolkit paper with useful contributions, particularly in extending fairness approaches to new domains like NLP and computer vision. The main concern is the limited technical novelty but the practical utility and engineering contributions outweigh this limitation for a toolkit paper.